# Notch and retinoic acid signals regulate macrophage formation from endocardium downstream of Nkx2-5

Norika Liu [1,2], Naofumi Kawahira[2], Yasuhiro Nakashima[3], Haruko Nakano[2], Akiyasu Iwase[4], Yasunobu Uchijima[4], Mei Wang[1], Sean M. Wu [5], Susumu Minamisawa[1], Hiroki Kurihara [4] & Atsushi Nakano[1,2,6,7,8] ✉

Hematopoietic progenitors are enriched in the endocardial cushion and contribute, in a Nkx2-5-dependent manner, to tissue macrophages required for the remodeling of cardiac valves and septa. However, little is known about the molecular mechanism of endocardial-hematopoietic transition. In the current study, we identified the regulatory network of endocardial hematopoiesis. Signal network analysis from scRNA-seq datasets revealed that genes in Notch and retinoic acid (RA) signaling are significantly downregulated in Nkx2-5-null endocardial cells. In vivo and ex vivo analyses validate that the Nkx2-5-Notch axis is essential for the generation of both hemogenic and cushion endocardial cells, and the suppression of RA signaling via Dhrs3 expression plays important roles in further differentiation into macrophages. Genetic ablation study revealed that these macrophages are essential in cardiac valve remodeling. In summary, the study demonstrates that the Nkx2-5/Notch/RA signaling plays a pivotal role in macrophage differentiation from hematopoietic progenitors.

The endocardium is a layer of endothelial cells with a transforming ability that line the innermost layer of the heart tube. Endocardial cells display remarkable plasticity during embryogenesis and give rise to the cushion endocardium, hematopoietic cells, endothelial cells in coronary and liver vasculatures, fibroblasts, mural cells, and adipocytes (summarized in a previous review article[1]). Recent reports suggest that a subset of endocardial cells gives rise to hematopoietic cells in an Nkx2-5-dependent manner during early embryogenesis[2–5]. The appearance of hemogenic endocardial cells spatiotemporally coincides with that of cushion endocardial cells, and they are both dependent on the expression of Nkx2-5, suggesting that they share common developmental programs[2,5,6]. This Nkx2-5-dependent cardiohematopoietic program is evolutionarily conserved among species as tinman (Drosophila homologue of Nkx2-5) is also required for both

cardiogenesis and hematopoiesis in the heart tube of Drosophila larva[7,8]. In addition, the major product of the Nkx2-5-dependent hematopoietic progenitors in the mouse heart is the macrophages as in tinman-dependent hematopoiesis in Drosophila.

Resident leukocytes are found in the developing heart as early as embryonic day (E) 10.5, even before the coronary and lymphatic vessels grow[9–11]. They play multiple roles during heart development, including coronary vessel formation[3,12] and lymphangiogenesis[9]. Genetic ablation of the endocardially derived macrophages results in cardiac valve remodeling[13]. This suggests that tissue-resident macrophages are essential for the formation of the local tissues during embryogenesis.

Although accumulating evidence suggests important roles for hematopoiesis in the heart tube, little is known regarding the genetic

[1]The Jikei University School of Medicine, Department of Cell Physiology, Tokyo, Japan. [2]University of California Los Angeles, Department of Molecular Cell and Developmental Biology, Los Angeles, USA. [3]Kyoto University, Department of Cardiovascular Medicine, Kyoto, Japan. [4]University of Tokyo, Department of Physiological Chemistry and Metabolism, Tokyo, Japan. [5]Stanford University, Cardiovascular Institute and Division of Cardiovascular Medicine, Department of Medicine, Stanford, USA. [6]University of California Los Angeles, David Geffen Department of Medicine, Division of Cardiology, Los Angeles, USA. [7]Eli and Edythe Broad Center of Regenerative Medicine and Stem Cell Research, University of California, Los Angeles, Los Angeles, USA. [8]Molecular Biology Institute, University of California, Los Angeles, Los Angeles, USA. ✉e-mail: anakano@ucla.edu

mechanisms of hematopoietic progenitors differentiate into the macrophages in the heart. During mammalian embryogenesis, Runx1 serves as a master regulator for definitive hematopoiesis in other established hematopoietic sites[14–17]. However, Runx1 knockout endocardium does produce CD41[+] hematopoietic progenitors and macrophages, suggesting that Runx1 is not required for the specification of macrophages[13].

In this study, we found that Notch and retinoic acid (RA) signaling is a critical regulator of endocardial hematopoiesis downstream of Nkx2-5. Analyses of single-cell RNA sequencing (scRNA-seq) datasets identified a distinct cluster of hematopoietic progenitors in heart tubes of mouse embryos at E9.5. This population was absent from Nkx2-5-null hearts, consistent with a previous report[2]. Further analyses of scRNA-seq data revealed that Notch signaling was significantly suppressed in Nkx2-5-null endocardial cells and that dehydrogenase/reductase 3 (Dhrs3), an NADPH-dependent reductase of retinaldehyde, was identified as a marker for Nkx2-5-dependent hematopoietic progenitors and macrophages. Genetic analysis revealed that enforced activation of Notch signaling resulted in a massive increase in CD41[+] progenitors in the endocardium and heart tube. Notch activation rescued the hematopoietic defect and cushion defect in Nkx2-5-null heart, suggesting that a Notch signal is downstream of Nkx2-5. Lineage tracing revealed that Dhrs3 is expressed specifically in the endocardial lining, hematopoietic progenitors and cardiac tissue macrophages derived from the Nkx2-5[+] endocardium in the cardiac cushion area, and Dhrs3[+] macrophages significantly increased in number by enforced activation of Notch signaling. Furthermore, ablation of Nkx2-5-derived macrophages resulted in defective cardiac valves, conforming with our previous finding using Nfatc1-derived tissue macrophage ablation[13]. Therefore, our study reveals that during development, endocardial hematopoiesis is regulated by the Nkx2-5-Notch and RA signals, which govern cardiac cushion remodeling via tissue macrophage differentiation.

## Results

### scRNA-seq analysis identified Nkx2-5-dependent hemogenic endocardial cells

To identify the molecular regulatory mechanism of Nkx2-5-dependent endocardial hematopoiesis, scRNA-seq data of E9.5 heart from wild-type (Ctrl) and Nkx2-5 knockout (KO) embryos obtained by SMART-seq2 on Fluidigm C1 system were analyzed (GSE76118; Fig. 1). A total of 289 cells were subjected to Louvain clustering, resulting in seven populations, including endocardial/endothelial cells (EC), four genetically distinct cardiomyocytes (CM), blood cells/leukocytes (Blood), and pro-epicardial cells (Epi) (Fig. 1b). On the basis of the expression profiles of four cardiomyocyte clusters, we annotated them as follows: ventricular-CM (Myl2[+]; cluster 1), Nkx2-5 independent-CM (only present in KO; cluster 2), atrial-CM (Itga6[+]/Nr2f2[+]; cluster 3), and outflow tract (OFT)-CM (Sema3c[+]/Bmp4[+]; cluster 4) (Supplementary Fig. 1a). Endocardial cells occupied nearly 40% of cells in Ctrl, and the population was reduced almost to half in KO hearts (Fig. 1c–e).

Endocardial/endothelial cells are heterogeneous not only in their origins but also in their functions[1, 18]. The EC cluster (cluster 0) was subdivided into five subclusters (Supplementary Fig. 1b) in Ctrl, but only three of them (0_0, 0_1, and 0_2) were present in KO (Fig. 1f, g). In Ctrl, the subcluster 0_0 represented the major population, while it was greatly reduced on KO cells. Hence, subcluster 0_2 became the major subpopulation in KO instead (Fig. 1g). Subcluster 0_3 and 0_4 were missing in the KO heart. Subcluster 0_3 was enriched for mesenchymal markers such as Twist1 and Msx1; thus, they were annotated as cushion endocardial cells (Fig. 1h). Subcluster 0_4 did not contain the pan-hematopoietic marker gene Ptprc (encoding CD45), but it expressed Pecam1, Runx1, and Itga2b (encoding CD41), suggesting that this subcluster likely corresponds to what we described CD31[+] Runx1[+] CD41[+] CD45[-] hemogenic endocardial cells that are connected with

surrounding endocardial cells via adherens junction (Fig. 1i, Supplementary Fig. 1c)[19,2]. This subcluster was distinct from circulating hematopoietic cells (cluster 5). This hematopoietic progenitors were also present in the heart tube in other 10X-based scRNA-seq data obtained from hearts at similar gestational stages (GSE126128; Supplementary Fig. 2).

Taken together, the scRNA-seq data successfully captured distinct populations of cushion endocardial cells and hematopoietic progenitors, both of which were absent from Nkx2-5-null hearts. These results conform well with endocardial phenotypes of Nkx2-5 knockout embryos known to date[2,6,20].

### Notch signaling regulates the formation of endocardial hematopoiesis and cushion downstream of Nkx2-5 transcriptional regulation

To identify the key signaling pathways regulated by Nkx2-5, Gene Ontology (GO) analysis was performed from differentially expressed genes (DEGs) in Ctrl and KO endocardial cells (cluster 0). The pathways related to phosphatidylinositol, small GTPase, Notch, TNF, and cyclic-nucleotide were significantly underrepresented in endocardial cells of Nkx2-5-null heart (Fig. 2a). Because the interplay between *tinman* (*Drosophila* homologue of Nkx2-5) and *notch* has been reported in *Drosophila* cardio-hematopoietic differentiation[7,8], we focused on Notch signaling pathway (GO: 0007219). Indeed, Notch-related genes, including Notch1, Jag1, Pln, and Psen2, were significantly downregulated in the KO endocardium (Fig. 2b and Supplementary Fig. 3a–c). A decrease in Notch activity in the Nkx2-5-null endocardium was also confirmed by immunostaining for Notch1 intracellular domain (NICD) (Fig. 2c and Supplementary Fig. 3d).

It has long been known that Notch signaling is required in hemangioblasts for the specification of hematopoietic cell fate in a cell autonomous manner[21–23]. To test whether Notch signaling also plays a role in the mammalian endocardial hematopoiesis, Notch signaling was forcibly activated in Nkx2-5[+] endocardial cells by crossing *R26[+/N1ICD-GFP]* mice with *Nkx2-5[cre/+]* mice[24]. Interestingly, embryos with conditional overexpression of N1ICD showed an accumulation of the red blood cells in the vasculatures of the head and the heart at E10.5 (white and black arrows in Fig. 2d(i)). Section staining showed a massive increase in CD41[+] hematopoietic progenitors in both the endocardium and circulation at E10.5 (Fig. 2d(i)–(iv)). Many of the CD41[+] cells in the endocardium were costained with GFP that labels N1ICD overexpressing cells (Fig. 2d(ii)). Flow cytometry analysis for CD31 and CD41 revealed that hematopoietic progenitors are significantly increased in N1ICD conditional overexpression at E10.0 (Fig. 2e and Supplementary Fig. 4). These data suggest that Notch activates the hematopoietic program in the Nkx2-5[+] endocardium.

Together, the in silico analysis and in vivo genetic manipulation suggest that Notch signaling regulates endocardial hematopoiesis in Nkx2-5[+] endocardial cells.

### Nkx2-5[+] cell-specific Notch signal activation in Nkx2-5null heart rescues endocardial cushion formation and endocardial hematopoiesis

To determine whether Notch signaling regulates endocardial hematopoiesis downstream of Nkx2-5, we conducted a genetic rescue experiment by activating a Notch signal in Nkx2-5-null background. As *Nkx2-5[+/cre]; R26[+/N1ICD-GFP]* mouse was embryonic lethal, *Nkx2-5[+/cre]* and *Nkx2-5[+/flox]; R26[+/N1ICD-GFP]* mice (both fertile) were crossed to obtain *Nkx2-5[Cre/flox]; R26[+/N1ICD-GFP]* "rescue" mice. We have confirmed that the phenotype of Nkx2-5[Cre/flox] embryos was identical to that of Nkx2-5-null embryos with cardiac looping defect, lack of mesenchymal transition of the endocardium (EndoMT), trabeculation defect, and lethality by E11. To avoid catastrophic defects, embryos at E9.5 were examined. At this stage, *Nkx2-5[Cre/flox]; R26[+/GFP]* "null" embryos had

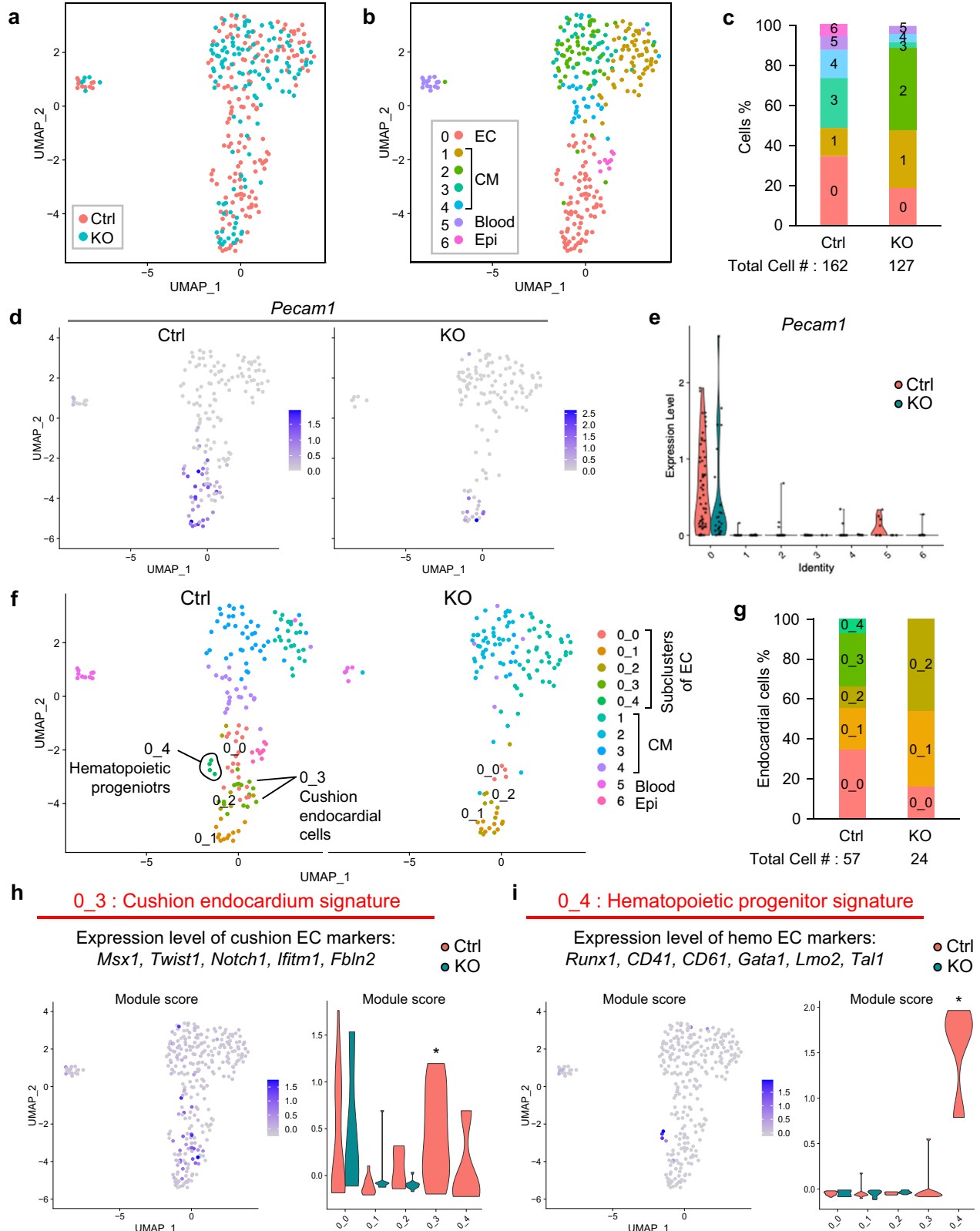

already begun to show pericardial effusion and anemia. Conversely, *Nkx2-5^Cre/flox; R26^+/N1ICD-GFP* "rescue" embryos showed no signs of these phenotypes, and the heart and dorsal aorta were replete with blood (Fig. 3a). Such blood-filled phenotype was seen in half (3 out of 6) of the *Nkx2-5^Cre/flox; R26^+/N1ICD-GFP* "rescue" embryos. The heart tube of the Nkx2-5-null embryo was shrunk because of the lack of cushion formation (yellow arrows in Fig. 3b(iii)), consistent with previous

reports that Notch signaling plays important roles in EndoMT during embryonic cardiogenesis[25–27]. However, forced expression of N1ICD restored the cushion formation in "rescue" embryos (white and yellow arrows in Fig. 3b(iv)). Moreover, the numbers of CD31⁺ CD41⁺ hematopoietic progenitors in endocardiac layer were both significantly restored by N1ICD overexpression (Fig. 3b(i)(ii)(i)'(ii)'). Quantification from section immunofluorescence staining showed

**Fig. 1 | scRNA-seq analysis identified Nkx2-5-dependent cushion endocardial cells and hematopoietic progenitors. a** UMAP representation of merged dataset with cells color-coded by genotype (Ctrl or KO). **b** UMAP representation of single-cell gene expression showing the seven identified clusters. EC: endocardial cell, CM: cardiomyocyte, Blood: blood cell (leukocyte), Epi: pro-epicardium (**c**) Stacked bar plot showing the relative proportion of clusters in each genotype, with total cell counts indicated. **d, e** UMAP feature plots and violin plot of *Pecam1* expression demonstrating that number of endocardial cells are decreased in KO. **f** UMAP representation of single-cell gene expression showing that the endocardial cluster

can be subclustered into five (0_0 to 0_4) in Ctrl, but only three subclusters appear in KO. **g** Stacked bar plot showing the relative proportion of clusters in each genotype, with total cell counts indicated. **h, i** Module scores projected on the UMAP and violin plots. The signature plot demonstrating that cluster 0_3 was enriched for cushion endocardium marker genes (*Msx1, Twist1, Notch1, Ifitm1,* and *Fbln2*) (H) and cluster 0_4 was enriched for hematopoietic progenitor marker genes (*Runx1, CD41(Itga2b), CD61 (Itgb3), Gata1, Lmo2,* and *Tal1*) (I). Cell clusters of interest are indicated by * in the violin plot.

that nearly 40% of N1ICD-induced endocardial cells (detected by GFP) in the rescue group expressed the CD41 hematopoietic progenitor marker (Fig. 3c). Thus, overexpression of N1ICD restored not only cushion formation but also hematopoietic endocardial cells. These data establish that the Nkx2-5-Notch axis plays a pivotal role in both EndoMT and endothelial-hematopoietic transition (EHT) in the endocardium.

### Retinoic acid signaling inhibits endocardial hematopoiesis via Dhrs3 expression

To further investigate the gene regulatory networks important in hematopoiesis in the endocardium, we used NicheNet, a program that predicts ligand–receptor and ligand–target links based on the scRNA-seq expression data and published protein-protein interactions[28]. All endocardium and cardiomyocyte clusters (clusters 0–4) were selected as potential sources of ligands and their corresponding receptors. Target genes in differentially expressed genes between Ctrl and KO were selected. These target genes were paired with their corresponding ligand genes according to the ligand–target matrix (Supplementary Fig. 5a). Ligand–receptor–target relationships (Supplementary Fig. 5a, b) were then established by selecting both receptors and target genes that are expressed in the hematopoietic progenitors (cluster 0_4) (Fig. 4a). Among the genes targeted by multiple ligands, Dehydrogenase/reductase 3 (Dhrs3) was the only gene specific to hematopoietic progenitors (Fig. 4b). Dhrs3 is a retinaldehyde reductase that prevents the formation of retinoic acid (RA) by catalyzing retinaldehyde to retinol (vitamin A; Fig. 4c). The scRNA-seq data showed specific expression of Dhrs3 in hematopoietic progenitors and low expression of cellular retinoic acid binding protein 2 (Crabp2) that promotes RA signaling by transporting RA to its receptor in the nucleus, implying that RA signaling is suppressed in hematopoietic progenitors in the endocardium (Fig. 4d).

To validate that Dhrs3 is expressed in hematopoietic progenitors in vivo, immunofluorescent staining analyses of Dhrs3, along with CD41 or CD31, were performed. Staining of E9.5 embryonic heart confirmed that Dhrs3 was expressed in CD41+ hematopoietic clusters on the endocardium (Fig. 4e) and that Dhrs3+ endocardial cells were absent in Nkx2-5 KO hearts in vivo (Fig. 4f). Approximately 1/3 of CD31+ endocardial cells were positive for Dhrs3, which was significantly higher than Nkx2-5-null hearts (Fig. 4g). Immunostaining for Dhrs3 and CD68 (a pan-macrophage marker) showed that Dhrs3 was also expressed in 33% of the macrophages in wild-type heart at E13.5. Overexpression of N1ICD increased not only the total number of CD68+ macrophages but also the Dhrs3+ fraction of macrophages in *Nkx2-5^cre/+; R26^+/N1ICD-GFP* mice (75%), higher than that in wild type (33%) (Fig. 4h). Of note, *Nkx2-5^cre/+; R26^+/N1ICD-GFP* mice were underrepresented at around E10, and only 1 out of 16 E13.5 embryos were obtained. Therefore, we reproduced the above results using *Nkx2-5^cre/+; CAG-CAT-N1ICD^tg* mice, which are not lethal at E13.5. *CAG-CAT-N1ICD^tg* mice have been reported to show appropriate activity in the heart but higher survival rate[29]. *Nkx2-5^cre/+; CAG-CAT-N1ICD^tg* mice also showed an increase in Dhrs3-positive endocardial cells and macrophages at E9.5 and E13.5, respectively (Supplementary Fig. 6a–c). Analysis of another set of scRNA-seq using embryonic and neonatal hearts confirmed the sustained expression

of Dhrs3 in endothelial cells and macrophages at E9.5, E11.5, E14.5, E18.5, P0, P7, and P21 (Supplementary Fig. 6d)[30].

Taken together, these data suggest that Dhrs3, a reductase that suppresses RA signaling, is a marker for Nkx2-5-dependent hematopoietic progenitors and that the activation of Nkx2-5-Notch increases the expression of Dhrs3 in macrophages in the heart.

### Notch signaling promotes whereas RA signaling inhibits hematopoiesis as well as macrophage differentiation

Although our analysis of the scRNA-seq data implies that RA signaling might be suppressed in the Nkx2-5-dependent hematopoietic progenitors, previous studies found mixed results as to the role of the RA signal in the hematopoietic activity in endothelial cells. Some reports demonstrated that RA is required for hematopoietic endothelial cell differentiation during embryogenesis[31–35], and others supported an inhibitory role of RA in hematopoiesis[36,37]. To examine whether the RA signal regulates hematopoiesis in the endocardium, we performed an ex vivo hematopoietic colony-forming assay with or without all-trans RA (atRA, a bioactive form of RA), 4-diethylaminobenzaldehyde (DEAB, an inhibitor of RA synthase), and DAPT (a γ-secretase inhibitor that blocks the Notch pathway). Optimal doses of atRA and DEAB were determined by analyzing mRNA level of retinoic acid receptor-β (RARβ) on day 4 of addition (Supplementary Fig. 7a). To examine whether Notch and RA signals regulate hematopoiesis at the specification phase, differentiation phase, or both, the chemicals were added in 3 regimens; (i) during OP9 culture, (ii) during methylcellulose culture, and (iii) both (Fig. 5a). When added during methylcellulose phase (regimen (ii)), the impact of atRA and DAPT is blunted and inconsistent. On the other hand, when added during the specification phase (regimen (i)), both atRA and DAPT significantly reduced the number of colony-forming units in all hematopoietic tissues (Fig. 5b). Treatment with both DEAB and DAPT resulted in decrease in the number of macrophages to the level of DAPT alone in regimen (i) (Fig. 5b). These trends were consistent in all 3 tissues. The presence of macrophages was confirmed by flow cytometry analysis of CD45+ Cx3cr1+ cells (Supplementary Fig. 7b).

These results indicate that macrophage differentiation from hematopoietic progenitor cells is promoted by Notch signaling and inhibited by RA signaling during the specification stages. The results of this in vitro experiment suggest that Notch signal regulates the hematopoiesis downstream of RA signal. However, scRNA-seq analysis and genetic manipulation of Nkx2-5 and Notch signal suggest that RA signal may be suppressed by Dhrs3 downstream of Notch signal (see Discussion).

### Macrophages derived from Nkx2-5+ endocardial cells are essential for the remodeling of the valves

We previously reported that endocardial-specific ablation of macrophages results in cushion remodeling defect in *Nfatc1^Cre/+; Csf1r^flox/flox* mice[13]. However, Nfatc1 labels entire endocardium and liver endothelium. Having found that Nkx2-5-Notch signaling promotes macrophage differentiation in the endocardium by inducing Dhrs3 expression (Fig. 4h), we next sought to investigate the role of macrophages generated from Nkx2-5-expressing endocardial cells. Nkx2-5-derived macrophages were ablated by generating *Nkx2-5^cre/+; Csf1r^flox/flox*

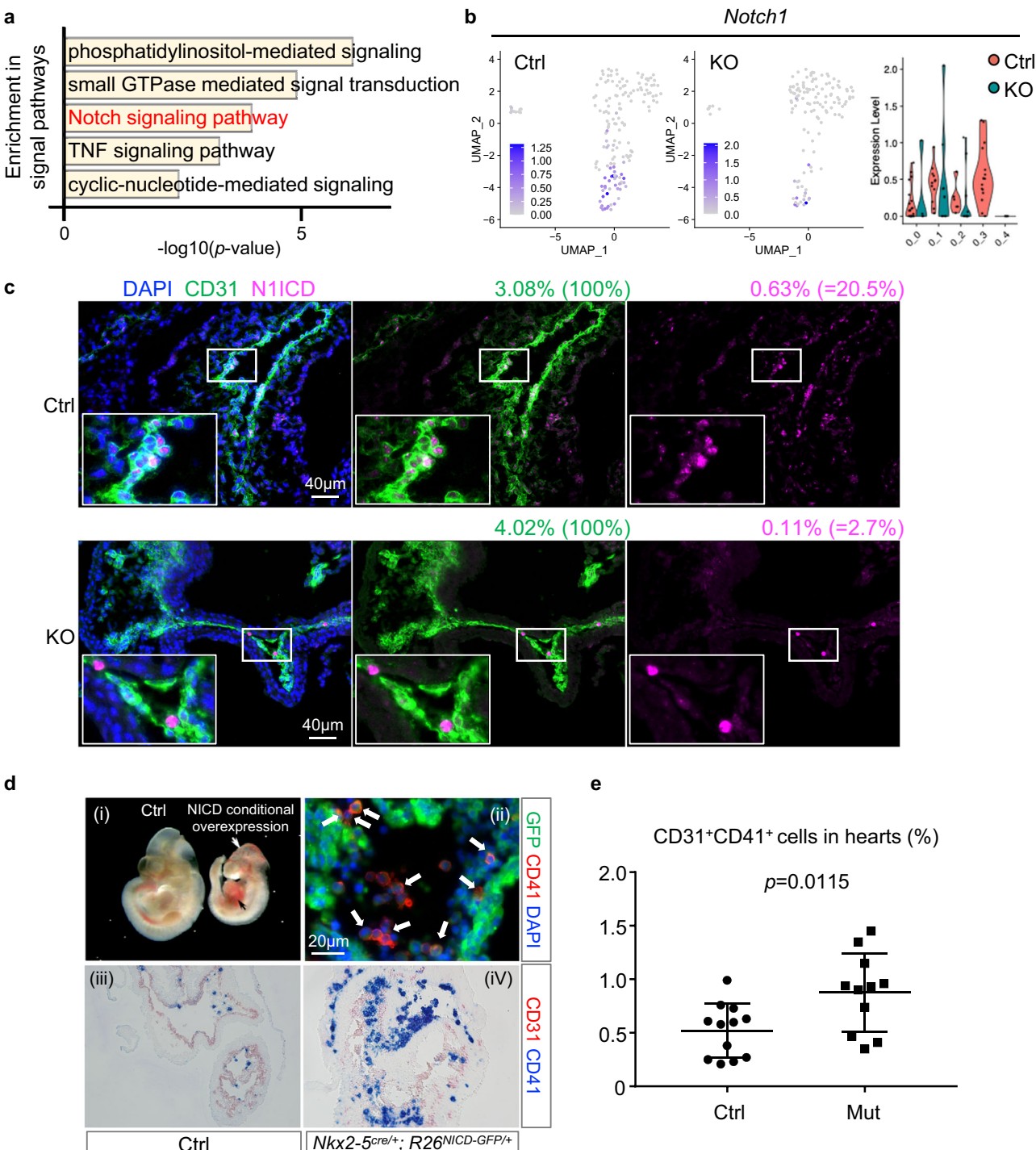

**Fig. 2 | Nkx2-5 regulation of Notch signaling is essential for endocardial hematopoiesis. a** Result from GO analysis from the differential expression genes (DEGs) between Ctrl and KO endocardium. **b** UMAP feature plots and violin plots showing the reduced expression level of *Notch1* in KO endocardium. **c** Cushion endocardium of Ctrl and KO embryos stained for N1ICD (pink), CD31 (green), and DAPI (blue). N1ICD was stained in the cushion endocardial cells in the control, but fewer in the Nkx2-5-null embryos. Five experiments were repeated independently with similar results. scale: 50 μm. **d** Overexpression of NICD in Nkx2-5 lineage cells activates hematopoiesis. (i) Ctrl and *Nkx2-5^{cre/+}; R26^{NICD-GFP/+}* embryos at E10.5. Although the mutants were smaller in body size, the size of the heart and the somite number were comparable. The heart was filled with blood (black arrow) and that head vasculatures are dilated and filled with blood as well (white arrow). (ii) Outflow

tract of *Nkx2-5^{cre/+}; R26^{NICD-GFP/+}* at E10.0. Many of the CD41+ cells in the outflow tract endocardium are costained with GFP reporter conjugated with N1ICD allele (white arrows). Many of the CD41+ cells are labeled with GFP reporter. (iii, iv) CD31 (red)/ CD41 (blue) staining of control (iii) and *Nkx2-5^{cre/+}; R26^{NICD-GFP/+}* (iv) embryos at E10.5 showed a striking increase in CD41+ cells (blue) in the mutant heart. **e** Flow cytometry analysis of the hearts from Ctrl and *Nkx2-5^{cre/+}; R26^{NICD-GFP/+}* (Mut) embryos at E10.0. The number of CD31 + CD41+ hematopoietic progenitors is significantly increased in the Mut hearts. Data represent mean ± SEM. Ctrl: 12 embryos were analyzed from 4 litters ($n = 12$), Mut: 11 embryos were analyzed from 3 litters ($n = 11$), $p = 0.0115$ from unpaired, two-tailed t-test. Source data are provided as a Source Data file.

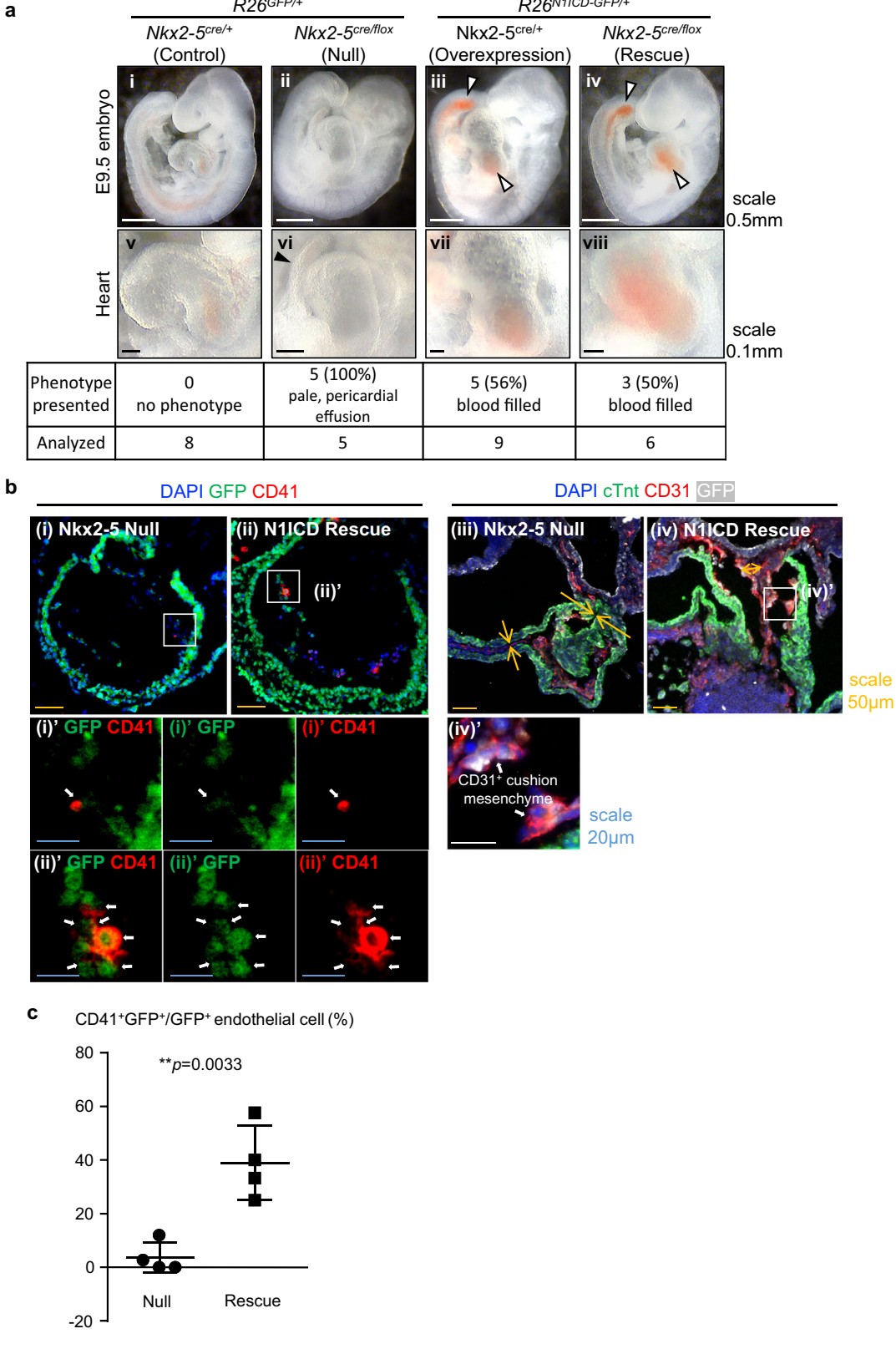

**c** CD41⁺GFP⁺/GFP⁺ endothelial cell (%)

mice. *Nkx2-5^(cre/+)*; *Csf1r^(flox/flox)* mutants were born at a normal Mendelian rate (Fig. 6a). However, mutant hearts were often enlarged (Fig. 6b). Histological analysis revealed thickened valves with occasional mucinous/myxomatous degeneration (Fig. 6c (arrow)). Furthermore, we observed that the valve thickening became evident from the early neonatal period (Supplementary Fig. 8a). Importantly, this phenotype is not due to mesenchymal cell proliferation but to excessive extracellular matrix deposition, as demonstrated by the accumulation of mucin (light blue) and/or collagen (yellow) by pentachrome staining and relative decrease in the cell density in the valve tissues (Supplementary Fig. 8b). These valvular phenotypes were consistent with our previous report with *Nfatc1-Cre* line[13]. The number of macrophages in the cardiac cushion and valve area was not statistically different between control and mutant animals (Supplementary Fig. 9) possibly

**Fig. 3 | Nkx2-5 mutant phenotype is rescued by overexpression of N1ICD.**
**a** Whole embryos (i-iv) and enlarged images of hearts (v-viii). Knockout embryo showed pericardial effusion (black arrow head). Caudal part of the dorsal aorta and heart of both Overexpression and Rescue embryos were filled with blood (white arrow head). The total number of embryos analyzed and the number of those that exhibited phenotypes are indicated in the table below. **b** Sections of ventricle of hearts of Null and Rescue embryos at E10.5. Nkx2-5-null endocardium rarely expressed CD41 ((i)(i)'). However, N1ICD overexpression (Rescue) restored endocardial CD41$^+$ cells, many of which are co-labeled with GFP ((ii)(ii)'). CD31 staining showed that Nkx2-5-null lacks cushion endocardium (iii), but the morphology of outflow tract of Rescue embryo appears normal with the presence of cushion endocardium (iv)(iv)'. **c** Quantification of CD41+ hematopoietic progenitors derived from Nkx2-5-lineage (GFP$^+$) by immunofluorescence imaging on null and rescue heart. Data are presented as mean values +/− SEM. Representative sections (those with the highest number of positive cells) from 4 embryos were quantified ($n$ = 4), $p$ = 0.0033 from unpaired, two-tailed t-test. Source data are provided as a Source Data file.

due to the compensation by tissue macrophages of other origins as previously reported[13,38,39]. It is unlikely that other immune cells contribute to this phenotype, as cells labeled with the T-cell markers, CD4, CD8, or the myeloid marker CD11b were rarely seen in the valve area in both control and *Nkx2-5$^{cre/+}$; Csf1r$^{flox/flox}$* mutants (Supplementary Fig. 10). These results suggest that Nkx2-5-lineage endocardium-derived macrophages are required for valve formation through their phagocytic role, consistent with a previous report[13].

## Discussion

Endocardial cells commit to multiple cell fates during embryonic development. After the onset of the heartbeat, endocardial cells begin to manifest their plasticity as they develop in cushion mesenchyme, coronary and liver vasculature, fibroblasts, mural cells, adipocytes, and hematopoietic cells (summarized in a previous study[1]). In this report, we investigated how the endocardial cells give rise to hematopoietic progenitors and macrophages during heart development. Using scRNA-seq analysis combined with in vivo and ex vivo validations, we identified that the Nkx2-5–Notch signaling axis is essential for the generation of both hematopoietic progenitors and cushion endocardial cells. We further found that Dhrs3, a suppressor of RA signal, is a marker of hematopoietic progenitors and their derived macrophages in the cardiac cushion. Functionally, Nkx2-5-derived macrophages played important roles in cardiac valve remodeling, consistent with our previous report using broader endocardial lineage-specific macrophage ablation[13].

Our analyses of the SMART-seq2-based scRNA-seq data of E9.5 Ctrl hearts revealed a cluster with both endothelial and hematopoietic makers ("hemogenic progenitors"; cluster 0_4) that was distinct from circulating hematopoietic cells (Fig. 1f, i). Consistent with the previous report[2], Nkx2-5 KO mice lacked the hematopoietic progenitor cluster without affecting the cluster of blood cells/leukocytes (Fig. 1f). Although Nkx2-5 is not specific to the endothelium of the heart[5,40], our scRNA-seq data clearly indicate that an endocardial cluster with hemogenic characteristics exists and that Nkx2-5 is essential for endocardial hematopoiesis but not for hematopoiesis from other anatomical sites. As reported previously, the majority of hematopoietic colonies from the heart give rise to macrophages and the macrophage formation in the endocardium is not Runx1-dependent[2,13]. In *Drosophila*, nearly 95% of blood cells in *Drosophila* are plasmatocytes, the equivalent of phagocytes and macrophages in mammals. The formation of plasmatocytes is known to be dependent on *tinman*, an Nkx2-5 homologue in *Drosophila*[7]. Our mouse studies revealed that hematopoiesis in the heart is macrophage-dominant and Nkx2-5-dependent (Fig. 6c). Thus, scRNA-seq revealed that the *Drosophila tinman*-dependent cardio-hematopoietic program is conserved in mammals.

Identification of the origin of the cells with mobile nature is often difficult. The limitations of existing endocardial lineage tracers have recently led to a debate on the presence of tissue macrophages of endocardial origin[40]. Since Nkx2-5$^+$ or Nfatc1$^+$ endothelial cells are also present in both the yolk sac and AGM[5,40], it is critical to demonstrate that the hematopoietic progenitors in the heart originate from the endocardium in situ and not from the yolk sac or AGM through circulation. Studies by us and others analyzing E9.5 embryos from

*Nfatc1$^{cre/+}$; R26$^{TtdTomato/+}$* demonstrated that the majority of endocardial cells were reporter positive, whereas the ratio of reporter-positive endothelial cells in the yolk sac and AGM were only marginal (less than 18%)[13,40]. However, analysis of E9.5 embryos in an inducible model using *Nfatc1$^{2A-CreER}$;R26$^{tdTomato}$* reporter mice treated with tamoxifen at E7.5 showed greater labeling of endothelial cells in the yolk sac and dorsal aorta (including AGM; approximately 45% and 32%, respectively)[40]. Given that Cre expression is reduced in the presence of IRES, it is possible that Nfatc1-IRES-cre line they used could not cover the putative hemogenic population in the endocardium. The current study revealed an Nkx2-5-dependent regulatory mechanism via scRNA-seq data analysis and supports the presence of Nkx2-5-dependent hematopoietic progenitors in the heart that differentiate into tissue macrophages (Figs. 1i and 4). Another limitation is that the data presented here do not necessarily exclude the non-cell-autonomous effect of Nkx2-5 and Notch. Genetic rescue experiments establish the epistatic relationship of Nkx2-5 and Notch signal in the endocardium (Fig. 3). However, it is also possible that the cushion endocardium serves as a niche for recruiting hematopoietic progenitors from outside of endocardium and that N1ICD overexpression primarily rescued cushion endocardium in *Nkx2-5$^{Cre/flox}$; R26$^{+/N1ICD-GFP}$* "rescue" embryos, thereby repopulating the non-endocardial Nkx2-5$^+$ progenitors to the cushion endocardium. Alternatively, as Nkx2-5 broadly labels the myocardium, ectopic N1ICD in the myocardium may non-cell-autonomously promote the hematopoiesis by recruiting the progenitors from outside of the endocardium. Studies are required to unequivocally prove that endocardial cells undergo hematopoietic transition.

In the current study, we identified that Notch signal drives the endocardial hematopoiesis downstream of Nkx2-5. We have shown that the Nkx2-5–Notch signaling axis plays important roles not only in the formation of hematopoietic cells and macrophages through EHT but also in the generation of the endocardial cushion mesenchyme via EndoMT (Figs. 2d and 3b). The role of Nkx2-5 and Notch signaling in these processes have been studied in the endocardium[2,3,5,25–27]. The novelty of our study is the establishment of a hierarchical relationship of Nkx2-5 and Notch in these two biologically important processes through genetic rescue experiment. Of note, genetic interaction between Nkx2-5 and Notch was also reported in *Drosophila* hematopoiesis[7] and mouse cardiogenesis[41]. Again, our results suggest that Nkx2-5-Notch axis is conserved in cardio-hematopoietic development across species.

The signal network analysis of scRNA-seq data (NicheNet) further identified Dhrs3 as a marker of hematopoietic progenitors in the heart and one of the earliest markers of cardiac tissue macrophages (Fig. 4). Dhrs3 was reported to be expressed in a small distinct population in the outflow cushion area of the developing heart[42], which is the primary site of apoptosis[43]. In the current study, we have discovered that this population corresponds to Dhrs3$^+$ tissue macrophages derived from the endocardium. The majority of cardiac tissue macrophages reside on the cardiac surface or within myocardium, and the unique localization patterns of the outflow cushion may indicate differences in macrophage origin. Although specific localization pattern and macrophage origins still remains to be elucidated, the fact that Nkx2-5-Notch signaling increased the number of Dhrs3$^+$ macrophages in the cardiac cushion region supports that endocardially-derived

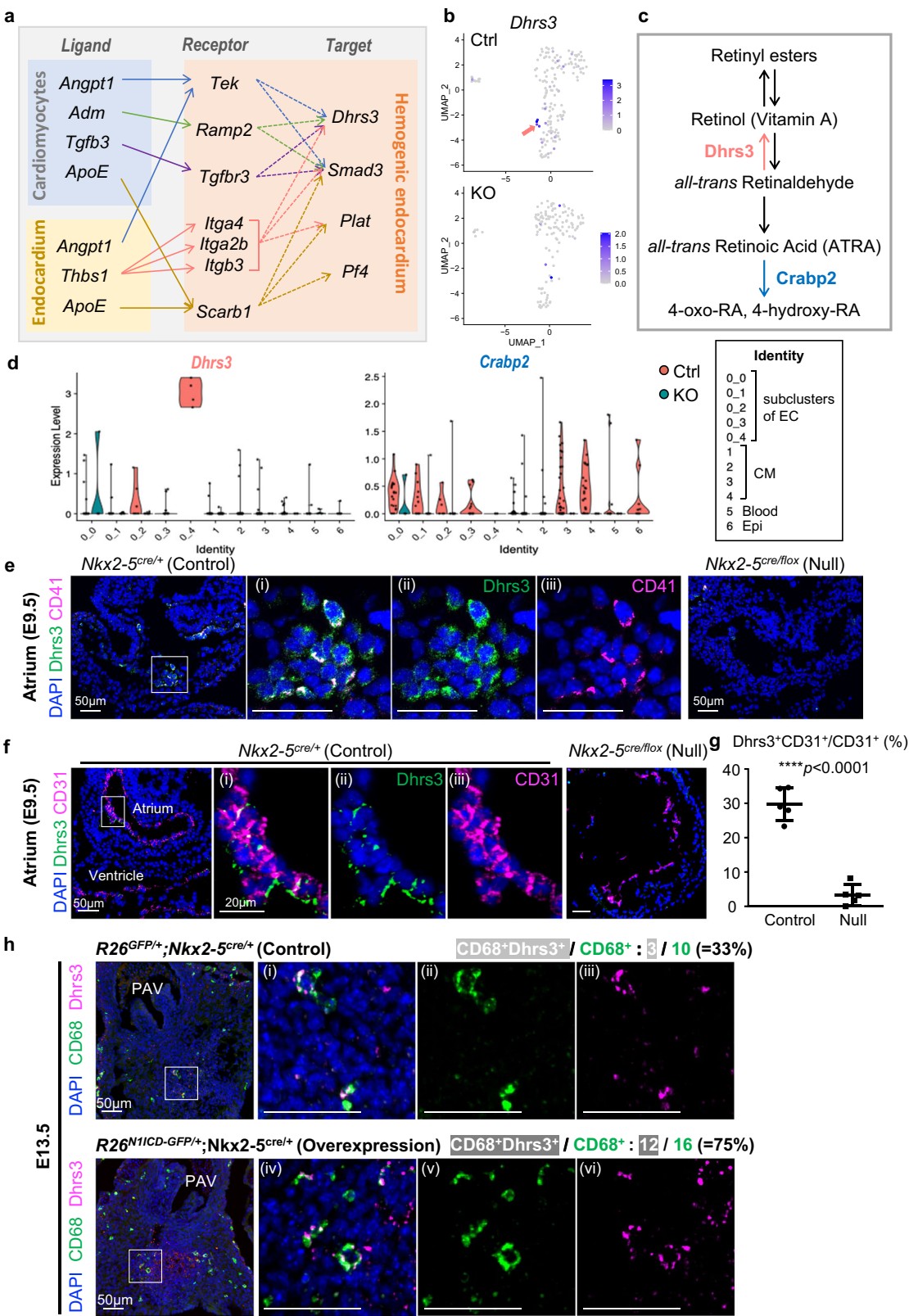

macrophages specifically contribute to this location (Fig. 4h and Supplementary Fig. 6b, c). The previous study analyzing the phenotype of Dhrs3 knockout mice identified multiple morphological abnormalities in the outflow tract region including double outlet right ventricle and ventricular septal defect[42], suggesting that these phenotypes were due to increased RA signaling caused by Dhrs3 deficiency. Although their study did not report a thickened phenotype of

the cardiac valves as we did in Nkx2-5-derived macrophage-ablated mice, it would be worth testing if Dhrs3 deficiency results in the ablation of macrophages and the valvular remodeling defect. If that were the case, cardiac valvular defects seen in Nkx2-5-derived macrophage-ablated mice could also be the result of an altered balance of RA signaling originating from reduced expression of Dhrs3. Thus, our study identified Dhrs3 as a marker of Nkx2-5-dependent

**Fig. 4 | Retinoic acid signaling is suppressed in endocardial hematopoiesis via Dhrs3 expression. a** Schematic presentation of ligand-receptor-target networks summarized from NicheNet analysis. Those with both receptors and targes are expressed in the hemogenic endocardium are shown. Arrows connecting ligands and receptors were predicted from the NicheNet packages as indicated in Supplementary Fig. 4c. Receptors and targets were connected (dashed arrows) based on the results of NicheNet ligand-target prediction as shown in Supplementary Fig. 4b. **b** UMAP feature plots showing *Dhrs3* expression is highly enriched in hemogenic EC in Ctrl but absent in KO. **c** Simplified scheme of RA metabolites. **d** Violin plots showed that *Dhrs3* is abundantly expressed and *Crabp2* is deficient in hemogenic EC (0_4). **e, f** Sections of atriums of control and Nkx2-5-null embryos at E9.5. Dhrs3 was specifically expressed on CD41⁺ cells, as well as CD31⁺ endocardial hematopoietic progenitors. Together with the results from our in vitro colony assay, its expression possibly plays a role in suppressing RA signaling to induce the formation of tissue macrophages essential for valve remodeling.

cells. **g** Dhrs3⁺CD31⁺ cells in total CD31⁺ cells were quantified from immunofluorescent staining. Data are presented as mean values +/− SEM. Representative sections (those with the highest number of positive cells) from 5 embryos were quantified (*n* = 5), *p* < 0.0001 from unpaired, two-tailed t-test. Source data are provided as a Source Data file. **h** Forced activation of Notch signaling increases the number of Dhrs3⁺ macrophages in the cardiac cushion at E13.5 embryos. Double positive cells for CD68 and Dhrs3 are indicated with yellow arrow heads. Ratio of Dhrs3⁺ macrophages were 33% and 75% in Control and Overexpression embryos, respectively. Only one embryos could be obtained, but the similar results were reproduced using *Nkx2-5^{cre/+}; CAG-CAT-N1ICD^{tg}* mice, which are not lethal at E13.5. These data are shown in Supplementary Fig. 6b, c. PAV Pulmonary arterial valve.

Regulation by RA signaling is highly context-dependent during development. Previous studies of EHT in the AGM region suggest that the specification of hemogenic endothelial cells requires activation of Notch signaling downstream of the RA[33]. In the absence of RA or Notch signaling, endothelial cells failed to commit to hemogenic fate, as demonstrated in retinaldehyde dehydrogenase 2 (Raldh2) knockout mice and DAPT-treated embryos[31–33]. Studies using in vitro differentiation of human pluripotent stem cells revealed mixed results. One study found that hematopoietic differentiation requires activation of RA signal[35], whereas the other found that RA signal rather needs to be suppressed[44]. Our data suggest that RA suppression is required for hematopoiesis as well as macrophage differentiation. The role of RA signaling in EHT may be dependent on the timing, dose of atRA, and/or differences between the primitive and definitive mechanisms of hematopoiesis. Although our study does not reveal a simple hierarchical relationship of Notch signaling and RA signaling, both in vivo and ex vivo examinations did show the interaction of two signaling mechanisms (Fig. 6d). Moreover, RA signaling is known to play essential roles not only in early embryonic hematopoiesis but also in heart morphogenesis[44–48]. In particular, it is well established that the formation of the outflow and inflow tracts is greatly influenced by excessive or deficient RA signaling[44–48]. Thus, the cardiac valve defects observed in Nkx2-5-derived macrophage-ablated mice could be due to abnormal RA levels within the outflow cardiac cushion region.

In summary, the current study demonstrated that Nkx2-5 is essential for the emergence of hematopoietic progenitor via the activation of Notch signaling and that the suppression of RA signaling in the heart. These results provide the evidence that Notch and RA signals play essential roles in regulating both EndoMT and hematopoietic differentiation downstream of Nkx2-5 in vivo.

## Methods
### Mice
All animal procedures conformed to the National Institutes of Health Guide for the Care and Use of Laboratory Animals and were approved by the Animal Research Committee at the Jikei University School of Medicine (approval no. 2020063) and UCLA Institutional Animal Care and Use Committee (protocol no. 2008-143). Mouse housing conditions are as follows: Lights go on at 7am and off at 7 pm. Humidity range is 45−55%. Temperature is maintained around 20−23 °C. All mouse strains were maintained in outbred backgrounds. Males and females were used in approximately equal numbers for all experiments. Nkx2-5-null embryos were generated by intercrossing *Nkx2-5^{cre/+}* mice[49]. For Nkx2-5-lineage reporter detection, *Nkx2-5^{cre/+}* was crossed with *Rosa26^{Tomato reporter/+}*. For the genetic rescue experiment, a Notch signal was conditionally activated in *Nkx2-5*-null background by

breeding *Nkx2-5^{cre/+}* and *Nkx2-5^{flox/+}; Rosa26^{N1ICD-IRES-GFP24}* to obtain *Nkx2-5^{cre/flox}; R26^{+/N1ICD-GFP}*, as *Nkx2-5^{cre/+}; R26^{+/N1ICD-GFP}* was embryonic lethal. *Nkx2-5*-lineage-specific N1ICD overexpression was generated by crossing *Nkx2-5^{cre/+}* and *CAG-CAT-N1ICD^{tg}* lines[29].

### Single-cell RNA sequencing (scRNA-seq) data analysis
scRNA-seq data (GSE76118) were analyzed with the R freeware (v4.1; R Core Team) with the Seurat package (v4.1). Seurat objects were created by choosing count matrix data for "e9.5_Nkx2-5wt" (=Ctrl) and "e9.5_Nkx2-5mut" (=KO). Data processing and normalization were performed according to the original publication[50]. Ctrl and KO datasets were integrated and analyzed using the FindIntegrationAnchors function of the Seurat package[51]. Cells in each cluster were annotated based on the results of FindAllMarkers. An endocardial cluster was further subclustered using the FindSubCluster function. The characteristics of endocardial subclusters were compared using the Find-Markers function. Identification of hemogenic and cushion endocardial cells was also performed on the GSE126128 data[52] using the FindSubCluster function described above. Signal pathway enrichment was analyzed from differentially expressed genes (DEGs; defined by FindAllMarkers function) in endocardial clusters between Ctrl and KO with Gene Ontology (GO) analysis using Metascape[53]. For NicheNet analysis, to include all possible signaling from either cardiomyocytes or endocardial cells, clusters 0, 2, 3, and 4 in Fig. 1b were included as both sender and receiver cells. By comparing the DEGs between Ctrl and KO, the ligands that had the most impact on the DEGs were prioritized. The NicheNet program was also used to predict the target genes and receptors of the prioritized ligands. An open-source R implementation of NicheNet is available at GitHub (https://github.com/saeyslab/nichenetr).

### Histological processing and immunostaining
Pregnant females were sacrificed at appropriate calculated embryonic ages by cervical dislocation following anesthetization with pentobarbital injection. Embryos were isolated in cold phosphate-buffered solution (PBS) and fixed in 4% paraformaldehyde (PFA) for the appropriate duration (E9.5 and E10.5: 30 min, E13.5: 2 h), followed by equilibration in 30% sucrose in PBS solution overnight. The samples were placed in 1:1 30% sucrose/OCT (Sakura Finetek, Electron Microscopy Sciences, Tokyo, Japan) solution for 1 h and then in 100% OCT compound for 1 h at 4 °C and finally embedded in 100% OCT carefully oriented in Cryomolds (Sakura Finetek). These blocks were cut in 10 μm thin serial sections with a Microm™ HM525 Cryostat (Thermo Fisher Scientific, Waltham, MA, USA).

For immunostaining, sections were briefly permeabilized with 0.2% Triton X-100/PBS and then blocked with 10% normal goad serum (NGS)/0.1% Tween-20/PBS (PBST). Primary antibody reactions were conducted in 5% NGS/PBST at 4 °C overnight. Secondary fluorescent conjugated antibody reactions were completed in 2% NGS/PBST for 1 h at room temperature. The primary antibodies used in this study were the following: rat anti-CD31 (1:200; BD Pharmingen, BD, Franklin Lakes, NJ, USA), rabbit-anti-activated Notch1 (1:100; Abcam, Cambridge,

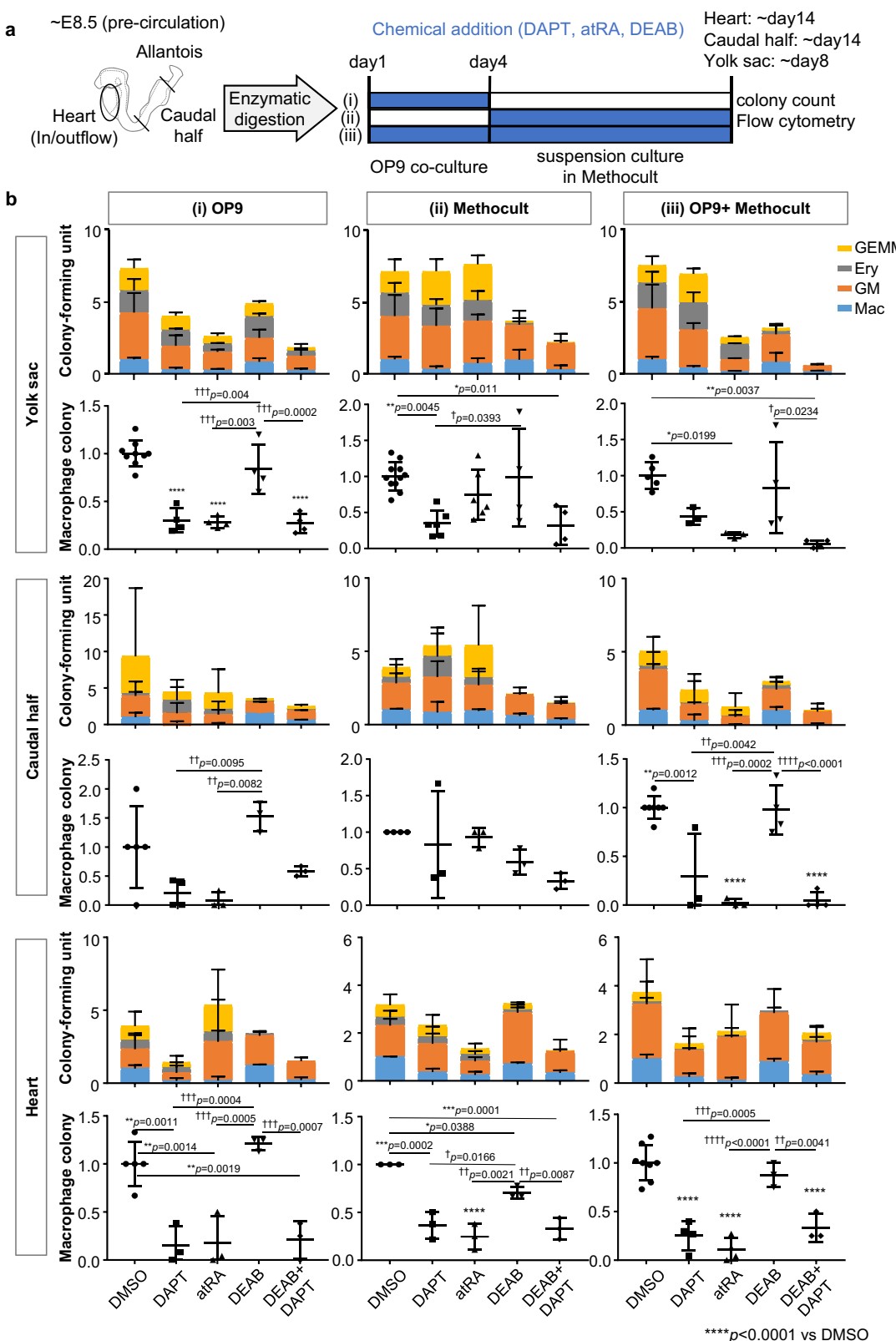

United Kingdom), rat anti-CD41 (1:200; BD Pharmingen), rat-anti-CD206 (1:200; BD Pharmingen), rat anti-CD68 (1:200; Bio-Rad Laboratories, Hercules, CA, USA), rabbit-anti-Dhrs3 (1:100; Proteintech, Rosemont, IL, USA), rabbit-anti-cTnT (1:250, Sigma-Aldrich, St. Louis, MO, USA), rabbit-anti-RFP (1:200; Proteintech), mouse-anti-GFP (1:200; Novus Biologicals, Centennial, CO, USA), and mouse-anti-MF20 (1:500; Hybridoma Bank). The secondary antibodies used in this study were the following: Alexa Fluor 488 goat anti- rat IgG(H + L) (1:1000), Alexa Fluor 488 goat anti- rabbit IgG(H + L) (1:1000), Alexa Fluor 555 goat anti- rat IgG(H + L) (1:1000), Alexa Fluor 594 goat anti-rabbit IgG(H + L) (1:1000), Alexa Fluor 647 goat anti- rat IgG(H + L) (1:1000), Alexa Fluor 647 goat anti- rabbit IgG(H + L) (1:1000), Alexa Fluor 647 goat anti- mouse IgG(H + L) (1:1000) (all from Invitrogen, Waltham, MA, USA). Slides were counterstained with DAPI (0.2 μg/mL;

**Fig. 5 | Notch signaling promotes, whereas retinoic acid suppresses, macrophage differentiation. a** Schematic representation of the three regimens of the colony-forming assay using hematopoietic organs of embryos at pre-circulation stage. Enzymatically digested tissues were co-cultured with OP9 feeder cells for 4 days, followed by methylcellulose culture with or without atRA (active RA), DAPT (Notch inhibitor) or DEAB (RA synthase inhibitor). Colonies were counted by the light microscopy and further analyzed by flow cytometry for macrophage ratio. **b** The number of hematopoietic colonies obtained from individual experiments was normalized by the number of macrophage colonies from the DMSO treatment. Results of the macrophage colony were pulled out in the lower panel of each tissue. Both DAPT and atRA consistently reduced hematopoietic activity in regimen (i), but

the effect was blunted and less consistent among tissue in regimen (ii). GM: granulocyte-macrophage, GEMM: granulocyte-erythroid-macrophage-megakaryocyte, Ery Erythrocyte, Mac Macrophage. Data are presented as mean values +/− SEM. Yolk sac: Four independent experiments with 1 or 2 replications in each condition were performed ($n = 4$). Caudal half: Four independent experiments (5 embryos were pooled for each experiment) in each condition were performed ($n = 4$). Heart: Three independent experiments (5 embryos were pooled for each experiment) with 1 or 2 replications in each condition were performed ($n = 3$). $p$ values were calculated by One-way ANOVA with post-hoc Tukey's multiple comparisons test. Source data are provided as a Source Data file.

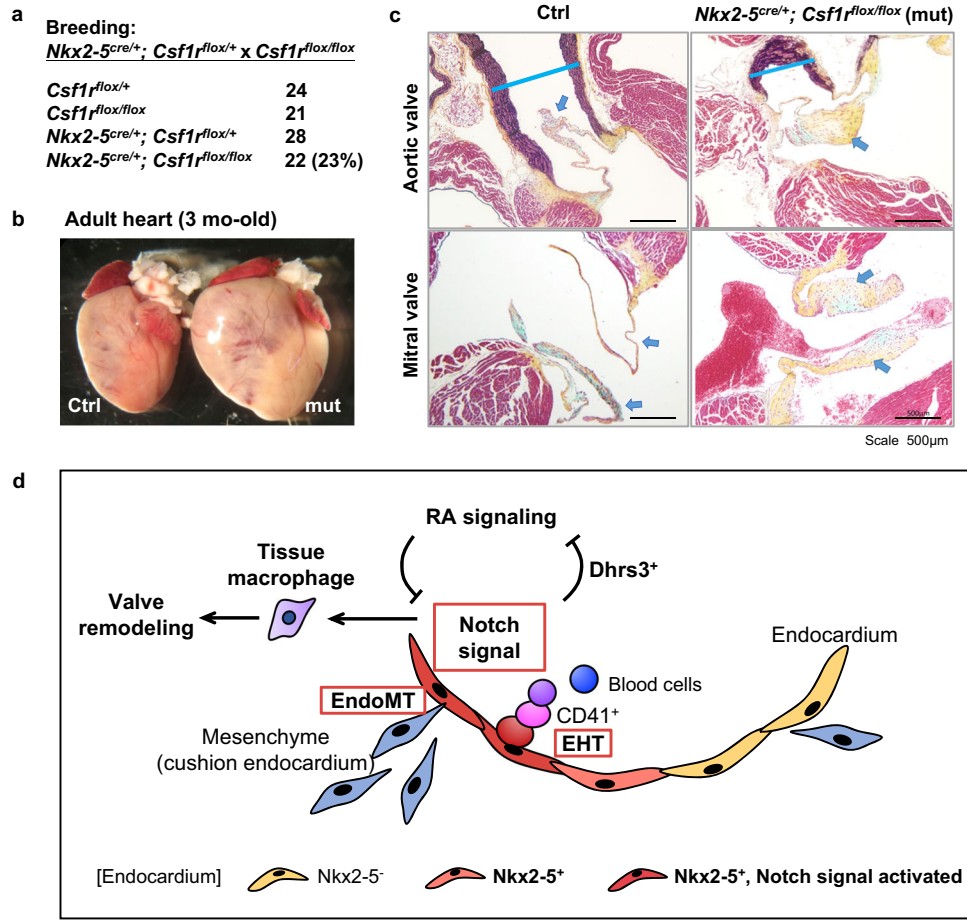

**Fig. 6 | Ablation of Csf1r (macrophage surface marker) in Nkx2-5 lineage results in cardiac enlargement, valvular degeneration, and narrow aortic root.** *Nkx2-5*<sup> cre/+</sup>*; Csf1r*<sup>flox/flox</sup> hearts and their controls (3 other genotypes) were dissected at 12 weeks old. Mice were all in C57Bl/6 background. **a** Mutants were born at normal Mendelian rate. **b** Appearance of whole heart. Note the enlargement of the heart. **c** Histology of aortic root and valves by Pentachrome staining. Valves show a

spectrum of degenerations (light blue arrow). Blue line indicates the aortic root measured just above the aortic cusp. Three experiments were repeated independently and two of them showed the similar results. **d** Schematic summary of the study. Notch signaling induces macrophage differentiation of Nkx2-5⁺ hemogenic endocardial cells by suppressing retinoic acid signaling through Dhrs3 expression. Macrophages of Nkx2-5-lineage are required for valve maturation.

Dojindo, Kumamoto, Japan) and mounted with SlowFade™ Gold Antifade Mountant (Invitrogen). Immunostaining procedures were followed by the manufactural instruction of M.O.M. kit (VECTOR Laboratories, Newark, CA, USA) when mouse primary antibodies were used.

For pentachrome staining, tissues were embedded into paraffin after fixing them in 10% formalin for overnight. Paraffin sections were made with 8μm thin with a microtome. The staining procedures were followed per manufacturer's instruction of Movat Pentachrome Stain Kit (ScyTek Loboratories, Logan, Utha, USA). Immunohistochemical staining were also performed using the serial paraffin sections indicated. Rabbit-anti-CD4 (1:500; Abcam), rat-anti-CD8 (1:500,; Novus Biologicals), and rat-anti-CD11b (1:500; BD Biosciences) were used as

primary antibodies and detected using Vectastain ABC-HRP Kit (VECTOR Laboratories).

Images were taken using a Zeiss LSM880 confocal microscope (Carl Zeiss, Oberkochen, Germany) or Keyence BZ-9000 (Osaka, Japan). Confocal microscope images were analyzed using ZEN 2.0 Black software (Carl Zeiss) and Imaris 8 software.

**Hematopoietic colony-forming assay**
The heart, yolk sac, and caudal half of the embryo, including the aorta-gonad-mesonephros (AGM) region, were harvested from E9.5 embryos and dissociated with an enzyme solution containing collagenase 2 mg/mL (CLS-2; Worthington, Lakewood, NJ, USA), dispase 0.25 mg/mL (17105−041; Gibco, Waltham, MA, USA), DNAase I (Invitrogen), 1%

penicillin/streptomycin (15140–122; Invitrogen) and 10% fetal bovine serum (Gibco) in PBS. The single-cell suspension was washed with 5% FBS and PBS, and then, cells were cultured on OP9 stromal cells for 4 days in 48-well plates in 500 mL of alpha-MEM (Gibco/Invitrogen) containing 20% FBS and 1% penicillin/streptomycin and supplemented with stem cell factor (SCF, 50 ng/mL), inter-leukin-3 (IL-3, 5 ng/mL), IL-6 (5 ng/mL), thrombopoietin (5 ng/mL) and Flt-3 ligand (Flt-3 L, 10 ng/mL) as previously described[2]. The cells were then dissociated mechanically from stroma with a transfer pipette and filtered to remove the remaining stromal cells (celltreck, Sysmex, Kobe, Japan). Filtered cells were transferred into 1.5 mL methylcellulose with SCF, IL-6, IL-3, and EPO (MethoCult 3434, STEMCELL Technologies, Vancouver, Canada) to determine multilineage potential. Colonies were scored 4–7 days later. DAPT (20 µM, FUJIFILM, Tokyo, Japan), all-trans-retinoic acid (0.1 µM, FUJIFILM), and diethylaminobenzaldehyde (10 µM, MedChemExpress, Monmouth Junction, NJ, USA) were added on the first day of OP9 coculture and methylcellulose culture. Dimethyl sulfoxide (DMSO) was used as vehicle. Hematopoietic colonies were observed using IX71 inverted system microscope (Olympus, Tokyo, Japan).

### Flow cytometry

Hematopoietic progenitors in hearts were detected with CD31 (FITC-conjugated, 1:200; BD Pharmingen) and CD41 (PE-conjugated, 1:200; Biolegend, San Diego, California, USA). Cells were collected from embryonic heart at E10.0 by dissociating tissues with the enzyme solution indicated above in Hematopoietic colony-forming assay section.

Hematopoietic colonies generated from the heart, yolk sac, and caudal half of the embryo, including the AGM region, were collected from E8.5 embryos and washed with 5% FBS and PBS to make the single-cell suspension. Samples were blocked with mouse Fc block (BD Biosciences) for 5 min, followed by labeling for 25 min at 4 °C with CD45 (APC-Cy7-conjugated, 1:200; BD Pharmingen) and Cx3cr1 (APC-conjugated, 1:200; Biolegend) antibodies for macrophage detection. Nonviable cells were excluded using 7-amino-actinomycin D (7-AAD, BD Biosciences). Cells were then analyzed by MACSQuant (Miltenyi Biotec, North Rhine-Westphalia, Germany) with the rat anti-monoclonal antibodies. All flow cytometry data analyses were done using FlowJo (version 7, BD Biosciences).

### Statistical analysis

All data were presented as mean ± standard error of the mean (SEM). Statistical analysis was performed on GraphPad Prism 7. One-way ANOVA, followed by *post hoc* Tukey's multiple comparisons test, was used to compare the number of macrophage colonies between groups (Fig. 5). Unpaired Student's *t*-test was used for comparison of control (Ctrl) and experimental groups. Quantification of imaging data was performed using Fiji (ImageJ2, Version: 2.9.0).

### Reporting summary

Further information on research design is available in the Nature Portfolio Reporting Summary linked to this article.

## Data availability

The scRNA-seq data that analyzed in this manuscript were obtained from the GEO database, accession number GSE76118, GSE126128, and GNomEx database under accession numbers 272 R, 274 R, 275-292 R, 439 R, and 440 R [https://b2b.hci.utah.edu/gnomex/experiments/overview?idLab=7]. Source data are provided with this paper.

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

## Acknowledgements

The authors thank the lab members of Nakano lab (UCLA) and Kurihara lab (University of Tokyo). Special thanks to Mr. Tatsumi Hasebe and Dr. Mai Minato-Kanai. The study was supported by R01HL127427 from NIH, and the Fund for Joint International Research (19K24689) and the Fund for Early-Career Scientists (21363933) from Japan Society for the Promotion of Science (JSPS KAKENHI).

## Author contributions

This study was conceptualized by N.L. and A.N. and performed by Y.N., H.N., M.W., and Y.U. under the supervision of S.M., H.K., and A.N.; The sequencing data were curated and analyzed by N.L., N.K., A.I., and S.M.W.; The manuscript was written by NL and AN with the edits of all co-authors.

## Competing interests

The authors declare no competing interests.
