## [Peer Review File · Nature Communications]

REVIEWER COMMENTS

Reviewer #1 (Remarks to the Author):

In this paper the authors show that the Nkx2-5 transcription factor functions upstream of NOTCH and RA to regulate the endocardial hematopoiesis that will give rise to macrophages contributing to endocardial cushion remodeling. In this context, Notch inhibits RA signaling and favours hemogenic endocardium differentiation towards macrophages, that will contribute to outflow tract cushion remodelling and valve sculpting. scRNA-seq identified a distinct cluster of hemogenic endocardial cells in of E9.5 hearts. This cell population was absent in Nkx2-5 deficient hearts, as previously shown. Notch signaling was suppressed in Nkx2-5 endocardial cells, and Dhhrs3 was identified as marker of Nkx2-5-dependent hemogenic endocardial cells and macrophages derived from Nkx2-5+ endocardium. Forced Notch activation rescues the hematopoietic and cushion defects in Nkx2-5-deficient hearts, suggesting that Notch functions downstream of Nkx2-5 in this process. Forced Notch activation resulted in a significant increase of CD41+ progenitors in heart tube endocardium. Dhhrs3 is expressed in the hemogenic endocardium and cardiac tissue macrophages derived from the Nkx2-5+ endocardium in the cushion area, and Dhhrs3+ macrophages significantly increase after forced Notch activation. Nkx2-5-derived macrophages ablation results in defective, hypercellularized cardiac valves.

The approach and methodology used through the MS is sound, although the quality of the results obtained is not equally good. The scRNA-seq of E9.5 control and Nkx2-5-deficient hearts is very informative and clearly reveals the lack of two cell populations in Nkx2-5 hearts: cushion endocardium and hemogenic endocardium. Notch is one of the underrepresented pathways, suggesting that Notch activity is reduced in Nkx2-5-deficient hearts. Figure 2C (a-d) shows immunofluorescence staining of N1ICD in control and Nkx2-5-deficient hearts. It is very difficult to see endocardial N1ICD expression in control hearts and in the mutant heart is impossible to see how is organized and much less to detect the endocardium. The quality of the pictures should be very much improved and their magnification increased. Likewise, Figure 2D(b-d) should be improved, and the contrast for panels 2c,d, increased. Also, the text on top of the panel 2d should be eliminated. Can the authors quantify and analyze statistically the data of Figure 2Cb,d and 2Db?

The rescue experiments shown in Figure 3A are very nice and clear, but the authors do not mention in the results, figure legend or Methods, how many embryos of each class do the examine and apply statistics. The stainings in Figure 3B are of poor quality and they should be improved and CD41- and CD31-positive cell quantified with respect to the rest of endothelial cells (Dapi-positive).

In Figure 4 the authors identify through elegant bioinformatics analysis Dhhrs3 as a hemogenic-endocardial-specific marker, and stain E9.5 atria sections from control and Nkx2-5 mutant hearts. The image quality of Figure 4E should be improved, and the magnification of the sections increased, as it is very difficult to see. Likewise, some quantification of Dhhrs3-positive endocardial cells should be provided. Similarly, it would be helpful if the authors could show higher magnification and brighter pictures of the stainings shown in Fig. 4F.

In Figure 6, the authors show the hypercellular adult aortic valves in Nkx2-5Cre/+;Csf1rflox/flox mutants. Can the authors measure mesenchyme proliferation in this model to establish when hypercellularization occurs?

Discussion. Minor point: In page 7 bottom, there is PMID in the text, the authors should add the proper reference in the MS.

In the Discussion, the authors claim that the novelty of their study is the “establishment of a hierarchical relationship of Nkx2-5 and Notch” ... Nkx2-5 is transcription factor and Notch is a receptor that is activated by various ligands, can the authors speculate about how does Nkx2-5 acts upon Notch signaling?

Reviewer #2 (Remarks to the Author):

Liu and coworkers addressed endocardial formation of macrophage immune cells and their role in valvular malformation. They found that in the absence of Nkx2-5, a master regulator of cardiac development, Notch and retinoic acid signalling together with downstream Dhhrs3 expression were abrogated, leading to reduced macrophage differentiation. This could be rescued by induction of Notch signaling in the Nkx2-5 lineage. Genetic ablation of macrophages in the Nkx2-5 was associated with cardiac developmental defects including valvular malformation. This analysis supports the role of endocardial macrophages in valve formation, and identifies novel molecular pathways in this pathology.

The authors present a timely and interesting study that follows their previous work but provides novel perspectives. The study is overall well-conducted with state-of-the-art techniques and lineage tracing models.

Comments:

The macrophage ablation study is not convincing. While this reviewer acknowledges that macrophages contribute to valve maturation, the phenotype displayed in figure 6 in Nkx2-5Cre;Csf1rflox/flox mice indicates a generally disturbed cardiac development that likely involves various signalling pathways. Given the massive cardiac enlargement, it is also difficult to differentiate primary and secondary valve defects. Thus, the specificity of this model needs to be defined. Considering the [proteinatlas.org](https://www.proteinatlas.org), Csf1r is also expressed on cardiac cells including myocytes and stromal cells (<https://www.proteinatlas.org/ENSG00000182578-CSF1R/tissue/heart+muscle>). Thus this conditional deletion model can be expected to provide macrophage-independent effects, which require further work up.

Does Csf1r inhibition results in a similar cardiac phenotype in embryonic mice?

In the setting of macrophage depletion, do the authors observe infiltration by other immune cells?

Runx1-independence of haematopoiesis can indeed allow the differentiation between primitive and definitive haematopoiesis. However, it should be kept in mind that requirements for Runx1 are

temporal, e.g. as demonstrated by the group of Nancy Speck (<https://pubmed.ncbi.nlm.nih.gov/23924635/>), and thus such analysis hugely depends on timing. And it is interesting to see that Runx1 is expressed in a subcluster that is absent in Nkx2-5 KO embryos.

Fig. S6: CD11b is not a macrophage marker but labels also other myeloid cells. Would be better to address F4/80, CD64 or Cx3cr1 (the latter is expressed earlier and identifies macrophage progenitors).

Reviewer #3 (Remarks to the Author):

This is an interesting and timely manuscript that aims at dissecting the pathways involved in macrophage specification from endocardial cells in the early embryo.

The approach is very elegant since the authors use a combination of single cell transcriptomics, functional genetic studies (including tissue-specific genetic rescues) and ex vivo hematopoietic assays using the mouse model. The main findings are the following: 1) the identification of a subset of Nkx2-5-dependent endocardial cells expressing hematopoietic markers; 2) the identification of Notch pathway as a critical positive regulator of macrophages found in the heart; 3) the identification of Retinoic Acid (RA) signaling as another regulator of cardiac macrophages development.

The question addressed is undoubtedly important and timely, as the functional identification of particular tissue-resident macrophage subsets will expand our understanding of tissue homeostasis and regeneration, a critical issue in particular in the heart field. In addition, a better characterization of the development of cardiac macrophage populations could, in principle, be exploited for the in vitro generation of these peculiar blood cell subsets that originates exclusively during embryonic development, for therapeutic purposes. However, in my opinion there are some limitations that are left unanswered and need to be addressed to grant a publication in Nature Communication.

According to this reviewer, there are two major issues:

1) Their definition of hemogenic endocardial cells is not accurate. The authors define hemogenic cells based on stainings for CD41 expression. Based on this, the cells they are looking at are not hemogenic but hematopoietic as they express already a hematopoietic marker. As such these cells cannot be defined as hemogenic (i.e. that can generate blood) as they are hematopoietic already.

In addition, there is no evidence that the hematopoietic cells observed emerge from bona fide endocardial cells. Indeed, their single-cell RNAseq analysis suggests that cells expressing hematopoietic genes are a separate cell population. There are certainly macrophages that emerge in a Nkx2-5-dependent manner and from Nkx2-5⁺ precursors, but it is entirely possible that Nkx2-5⁺ mesodermal cells are heterogeneous and can generate hemogenic endothelial cells that are distinct from endocardial cells. In fact, the data showed with their elegant genetic rescue of Notch signaling is compatible with alternative interpretations, i.e. that Notch signaling is required cell non autonomously for the emergence of cardiac macrophages in the early embryo, with the cushion endocardium acting as a niche

for the specification and development of these macrophages. In the absence of formal proof, this reviewer suggests replacing “hemogenic endocardial cells” with “Nkx2-5-dependent hematopoietic progenitors”.

2) The experimental design and therefore the results of the entire figure 5 are difficult to interpret. The ex vivo explant cultures is performed with tissues harvested from E8.5 embryos. An 18-day-long culture of heart explants in presence of the chemical Notch inhibition result in a decreased macrophage output compared to a vehicle-treated condition. Given the identification of Notch signaling as a critical regulator of Nkx2-5-dependent hematopoiesis, the authors use these results obtained with the chemical inhibition of Notch signaling as a validation of the platform. However, the timing of the genetic experiments show a full rescue at 9.5, which is compatible with a much earlier effect of Notch signaling. Hence, the results obtained by this ex vivo cultures raise the obvious question of the timing of Notch-dependence. When is then Notch needed for the Nkx2-5-dependent hematopoiesis? Is there a difference in Notch-dependency timing for setting up a cushion endocardial vs an hematopoietic program?

The RA part of the experiment is also very difficult to interpret. On one end for the timing, as again an 18-day-long culture in presence of RA agonist and antagonist is used while in vivo there are already Dhhrs3+ CD41+ positive cells. As per Notch, when RA is important? For the specification of the hematopoietic program? For the emergence of CD41+ progenitors? For their differentiation into macrophages? Both Notch and RA are highly dynamic signaling, a better characterization of the timing of the requirement of these signaling is needed to support the claims made by the authors. In addition, regarding the RA ex vivo studies, these are performed in presence of serum, which usually already contain retinol. Since the amount of atRA dictates the functional output (a recent Nat Cell Biol paper from the Sturgeon group show this elegantly in the context of hematopoietic development, and should probably be referenced as well), it would be important to test different doses of both the agonist atRA and antagonist DEAB. In addition, a proof that these small molecules are acting as expected (such as target genes expression) is totally lacking and very much needed.

Minor comments:

- Line 97 change “alienated” with “separated” or other synonym.
- In figure 5, many statistical analysis are missing, it is not clear whether they were not performed or whether the result are not significant.

Response to Reviewer #1:

General Comments

In this paper the authors show that the Nkx2-5 transcription factor functions upstream of NOTCH and RA to regulate the endocardial hematopoiesis

The approach and methodology used through the MS is sound, although the quality of the results obtained is not equally good.

➔ Reviewer #1 appreciates the sound approach but raises a concern about the quality of the images and the paucity of quantitative data. We thank Reviewer #1 for the thorough review of the manuscript. We revised our manuscript by taking into account all of the important comments. A large number of additional experiments are now presented in a total of 103 new panels with improved quality. Please find point-by-point response to the comments as follows. The reviewer's comments are shown in italicized font and our response is shown in black regular font.

Comments to Authors

Major Comments:

1. *Figure 2C (a-d) shows immunofluorescence staining of NIICD in control and Nkx2-5-deficient hearts. It is very difficult to see endocardial NIICD expression in control hearts and in the mutant heart is impossible to see how is organized and much less to detect the endocardium. The quality of the pictures should be very much improved and their magnification increased.*

➔ The experiment was repeated with freshly dissected embryos, and the representative pictures in Figure 2C are replaced with clearer images. The higher magnifications now enable readers to distinguish the cytosolic expression of *NIICD*. Quantifications showing the ratio of *NIICD* positive endocardial cells are provided (Ctrl: 20.5% vs KO: 2.7%).

Figure 2C

2. *Figure 2D(b-d) should be improved, and the contrast for panels 2c,d, increased. Also, the text on top of the panel 2d should be eliminated. Can the authors quantify and analyze statistically the data of Figure 2Cb,d and 2Db?*

➡ Cleaner images are provided in the revised manuscript. In addition, CD41⁺/GFP⁺ cells were quantified by flow cytometry (Figure 2E). Representative gating strategy for flow cytometry data were also provided in new Figure S4.

3. *The rescue experiments shown in Figure 3A are very nice and clear, but the authors do not mention in the results, figure legend or Methods, how many embryos of each class do the examine and apply statistics.*

➡ We thank the reviewer for pointing out the lack of critical information in the key data. In the revised version, we included the total number of embryos analyzed, the number of those that exhibited phenotypes, and a brief description of phenotype.

4. *The stainings in Figure 3B are of poor quality and they should be improved and CD41- and CD31-positive cell quantified with respect to the rest of endothelial cells (Dapi-positive).*

➡ The experiment was repeated and the images were re-taken as presented in new Figure 3B. GFP⁺ cells represent cells in Nkx2-5-lineage that are forcibly expressed N1ICD. Nkx2-5-null endocardium rarely expressed CD41 ((a)(a)'). However, NICD overexpression (Rescue) restored endocardial CD41⁺ cells co-labeled with GFP (Nkx2-5 lineage) ((b)(b)'). The number of CD41⁺ GFP⁺ hematopoietic progenitors are quantified from immunofluorescent stained images in Figure 3C.

Figure 3B

Figure 3C

5. *The image quality of Figure 4E should be improved, and the magnification of the sections increased, as it is very difficult to see. Likewise, some quantification of Dhhrs3-positive endocardial cells should be provided.*

➔ The experiments were repeated and we now have a magnified pictures presented in Figure 4E of the revised version (Figure 4Ea, b, c). In addition, we have added a representative image of Dhhrs3 and CD31 co-immunofluorescent staining (Figure 4F) and a quantification from the images (Figure 4G). Dhhrs3 expression in the endocardium was absent in Nkx2-5-null hearts (Figure 4E, F, G).

Figure 4E

Figure 4F

Figure 4G

6. Similarly, it would be helpful if the authors could show higher magnification and brighter pictures of the stainings shown in Fig. 4F.

➔ We replaced the enlarged images in Figure 4F in the revised version with a brighter, more enlarged images (Figure 4Ha-f).

Figure 4H

7. In Figure 6, the authors show the hypercellular adult aortic valves in *Nkx2-5Cre/+;Csf1r/flox/flox* mutants. Can the authors measure mesenchyme proliferation in this model to establish when hypercellularization occurs?

➔ We appreciate this suggestion. Our previous report using *Nfatc1-Cre;Csf1r-flox/flox* mice

(endocardially-derived macrophage ablation mice) also showed similar valvular defects (Shigeta et al., *Dev Cell*, 2019). In that study, we used PH3 antibody and found no difference in mitotic activity of the mesenchymal cells between control and mutant hearts (Figure S5 in Shigeta et al., *Dev Cell*, 2019). In the revised manuscript, we conducted a new set of experiments: aortic valves and mitral valves of *Nkx2-5Cre/+;Csflr-flox/flox* mutants at different stages (E15.5, P1, P10) were analyzed for ECM deposition by the Movat's pentachrome staining and cellular density by cell counts per area. As shown in Figures S8A and B, mutant valves are thicker but rather less cellularized, suggesting more ECM deposition. Therefore, the thickened phenotype of the valve leaflets is due to a tissue remodeling defect rather than an increase in cell proliferation. Accordingly, we replaced “hypercellularized valve” to “thickened valve” in the revised manuscript.

Figure S8A

Figure S8B

Minor Comments:

1. In page 7 bottom, there is PMID in the text, the authors should add the proper reference in the MS.

➔ We have put the proper reference. We thank the reviewer for pointing it out.

2. In the Discussion, the authors claim that the novelty of their study is the “establishment of a hierarchical relationship of Nkx2-5 and Notch”... Nkx2-5 is transcription factor and Notch is a receptor that is activated by various ligands, can the authors speculate about how does Nkx2-5 acts upon Notch signaling?

➔ As shown below, the enhancer-promoter searches using consensus sequence data and H3K4me1- and H3K27ac-ChIP sequence data from embryonic heart have revealed Nkx2-5 binding sites in the Notch signaling receptor (Notch1) and ligand (Jag1). These consensus sequences are conserved among species and close to the binding sites for other hematopoietic and cardiac transcription factors including GATAs and TBXs. Therefore, although it still remains speculative, it is likely that at least some components of Notch signaling are direct targets of Nkx2-5.

Response to Reviewer #2:

General Comments

Liu and coworkers addressed endocardial formation of This analysis supports the role of endocardial macrophages in valve formation, and identifies novel molecular pathways in this pathology.

The authors present a timely and interesting study that follows their previous work but provides novel perspectives. The study is overall well-conducted with state-of-the-art techniques and lineage tracing models.

➔ Reviewer #2 appreciates the significance and novelty of this study, and we feel honored to receive such positive comments. We agree to all the comments and criticisms. In the revised manuscript, the key issues are amended based on the suggestions from Reviewer #2 mostly by conducting additional experiments and a few by amending the discussion. Please find point-by-point response as follows.

Comments to Authors

Major Comments:

1. *The macrophage ablation study is not convincing. While this reviewer acknowledges that macrophages contribute to valve maturation, the phenotype displayed in figure 6 in *Nkx2-5Cre;Csf1r-flox/flox* mice indicates a generally disturbed cardiac development that likely involves various signalling pathways. Given the massive cardiac enlargement, it is also difficult to differentiate primary and secondary valve defects. Thus, the specificity of this model needs to be defined. Thus this conditional deletion model can be expected to provide macrophage-independent effects, which require further work up. Does *Csf1r* inhibition results in a similar cardiac phenotype in embryonic mice?*

➔ In a severely affected tissue, it is often difficult to dissect the cause and the consequence without examining the time course. In order to clarify if the valve defect is primary or secondary, we analyzed the valve morphology in the earlier stages including embryonic stage (E15.5, P1, and P10). The results showed that the aortic and mitral valves in the mutants were thickened from the neonatal stages, when the heart chamber was not enlarged yet (Figure S8A). The valve thickening was less prominent at the embryonic stage, likely because the valve remodeling and the degradation of extracellular matrix (ECM) are more drastic at neonatal stages. The ECM accumulation was analyzed by calculating the percentage of nuclei to valve leaf area. This new set of experiments revealed that, at P1 and P10, the density of nuclei was significantly reduced in mutants, and the deposition of ECM was increased. $n=4$, $**p<0.01$, $***p<0.001$. t-test between Ctrl and Mut at respective stages (Figure S8B). Thus, the valve phenotype precedes the enlargement of the heart, suggesting that valve remodeling defect is the primary phenotype.

Figure S8A

Figure S8B

Figure S9

2. *In the setting of macrophage depletion, do the authors observe infiltration by other immune cells?*

➡ To address this question, we performed immunohistochemical staining of CD4, CD8, and CD11b to detect T cells and myeloid cells in P10 valves, when the valves show more deposition of collagen compared to the Ctrl. The cells positive for CD4, CD8, and CD11b were rarely seen in the valve area and no significant change in their numbers were found (Figure S9). Therefore, the contribution of other immune cells to the enlarged valve is minimal and there is no noticeable increase in the mutants.

3. *Runx1*-independence of haematopoiesis can indeed allow the differentiation between primitive and definitive haematopoiesis. However, it should be kept in mind that requirements for *Runx1* are temporal, e.g. as demonstrated by the group of Nancy Speck (<https://pubmed.ncbi.nlm.nih.gov/23924635/>), and thus such analysis hugely depends on timing. And it is interesting to see that *Runx1* is expressed in a subcluster that is absent in *Nkx2-5* KO embryos.

➡ We thank the Reviewer #2 for this insightful comment. We agree that it is important to note that *Runx1* requirement for hematopoiesis is only temporal. Our scRNA-seq data suggest that *Runx1* expression in hemogenic endocardial cells is dependent on *Nkx2-5*, and our published data suggest that *Runx1* is dispensable for macrophage formation in the endocardium (Shigeta, *Dev Cell*, 2019). These two observations led to our hypothesis that macrophages are formed via a primitive mechanism, although it is beyond the scope of this manuscript to prove it.

4. *Fig. S6: CD11b is not a macrophage marker but labels also other myeloid cells. Would be better to address F4/80, CD64 or Cx3cr1 (the latter is expressed earlier and identifies macrophage progenitors).*

➡ Thank you for pointing out our oversight. *CD11b* is a marker for myeloid cells, and not specific to macrophages. We have repeated a series of hematopoietic colony assay and re-analyzed by flow cytometry using *Cx3cr1* in the revised manuscript (Figure S7B). The rapid expansion in the in vitro system has blunted the change in the number of macrophages but the conclusion was not changed.

Figure S7B

Response to Reviewer #3:

General Comments

This is an interesting and timely manuscript that aims at dissecting the pathways involved in macrophage specification from endocardial cells in the early embryo.

The approach is very elegant since the authors use a combination of single cell transcriptomics, functional genetic studies (including tissue-specific genetic rescues) and ex vivo hematopoietic assays using the mouse model. The main findings are the following: 1) the identification of a subset of Nkx2-5-dependent endocardial cells expressing hematopoietic markers; 2) the identification of Notch pathway as a critical positive regulator of macrophages found in the heart; 3) the identification of Retinoic Acid (RA) signaling as another regulator of cardiac macrophages development.

The question addressed is undoubtedly important and timely, as the functional identification of particular tissue-resident macrophage subsets will expand our understanding of tissue homeostasis and regeneration, a critical issue in particular in the heart field. In addition, a better characterization of the development of cardiac macrophage populations could, in principle, be exploited for the in vitro generation of these peculiar blood cell subsets that originates exclusively during embryonic development, for therapeutic purposes. However, in my opinion there are some limitations that are left unanswered and need to be addressed to grant a publication in Nature Communication.

➡ We thank Reviewer #3 for the fair and rigorous analysis of our manuscript and the insightful comments as to the interpretation of the results. In particular, the points 1 and 2 are of fundamental importance to the concept of the hematopoiesis in the heart tube. Although we cannot solve this big question in this manuscript, we hope it will bring us closer to the identification of smoking gun. The point 3 is also a very important point which we overlooked in the first submission. These critiques have significantly improved the rigor of the manuscript, and we thank the Reviewer #3 for that. Please find our point-by-point response as follows.

Comments to Authors

Major Comments:

- 1. Their definition of hemogenic endocardial cells is not accurate. The authors define hemogenic cells based on stainings for CD41 expression. Based on this, the cells they are looking at are not hemogenic but hematopoietic as they express already a hematopoietic marker.*

➡ We thank Reviewer #3 for this insightful comment regarding the definition of hemogenic endothelium/endocardium. Zhu et al defined hemogenic endothelial cells (HEs) as CD31⁺ Runx1⁺ CD45⁻ and intra-arterial clusters (IACs) as Runx1⁺ CD45⁺ (see the Figures below taken from Zhu, ..., Speck, Tan et al, Blood 2020). There is no mention about CD41 expression in Zhu et al paper. Our subcluster 1_4 is consistent with HEs in that it is CD31⁺ Runx1⁺ CD45⁻. However, our population does express CD41⁺ which may or may not be expressed in HEs defined by Zhu et al. As observed under the electron microscopy in our previous paper (Nakano et al, Nat Comm, 2013),

our CD31⁺ Runx1⁺ CD41⁺ CD45⁻ cell population resides within the endocardial layer and form adherens junction with surrounding endocardial. Therefore, we concluded that our population does not likely originate from circulation, making us use the term “hemogenic endocardium” in our previous publications. However, the reviewer’s point is well taken. In the revised manuscript, we refrain from using the word “hemogenic endocardial cells” and changed it to “hematopoietic progenitors”. We also added Figure S1C to show the expression profile of CD31⁺ Runx1⁺ CD41⁻ and CD45⁻.

Figure S1C (this manuscript), Endocardium

Figure 3A, B from Zhu et al., AGM region

2. *There is no evidence that the hematopoietic cells observed emerge from bona fide endocardial cells. it is entirely possible that Nkx2-5+ mesodermal cells are heterogeneous and can generate hemogenic endothelial cells that are distinct from endocardial cells. In fact, the data showed with their elegant genetic rescue of Notch signaling is compatible with alternative interpretations, i.e. that Notch signaling is required cell non autonomously for the emergence of cardiac macrophages in the early embryo, with the cushion endocardium acting as a niche for the specification and development of these macrophages. In the absence of formal proof, this reviewer suggests replacing “hemogenic endocardial cells” with “Nkx2-5-dependent hematopoietic progenitors”.*
 - ➡ We agree that we cannot exclude the possibility that the cushion endocardium serves as a niche for recruiting hematopoietic progenitors from outside of endocardium and NIICD overexpression primarily rescued cushion endocardium in *Nkx2-5^{Cre/flox}; R26^{+ /NIICD-GFP}* “rescue” embryos, thereby

repopulating the non-endocardial Nkx2-5⁺ progenitors to the cushion endocardium. Alternatively, as Nkx2-5 broadly labels the myocardium, ectopic N1ICD in the myocardium non-cell-autonomously promotes the hematopoiesis by recruiting the progenitors from outside of the endocardium. This point and the point 1 are both related to the origin of these hematopoietic progenitors. Ultimately, we will need time-lapse observation to unequivocally demonstrate the existence of “hemogenic endocardial cells”. Given the limitations of the current experimental systems, we agree to use the word “Nkx2-5-dependent hematopoietic progenitors”. This point is now discussed in the Discussion in the revised manuscript as follows: “Genetic rescue experiments establish the epistatic relationship of Nkx2-5 and Notch signal in the endocardium (Figure 3). However, we cannot exclude the possibility that the cushion endocardium serves as a niche for recruiting hematopoietic progenitors from outside of endocardium and N1ICD overexpression primarily rescued cushion endocardium in *Nkx2-5^{Cre/flox}; R26^{+N1ICD-GFP}* “rescue” embryos, thereby repopulating the non-endocardial Nkx2-5⁺ progenitors to the cushion endocardium. Alternatively, as Nkx2-5 broadly labels the myocardium, ectopic N1ICD in the myocardium non-cell-autonomously promotes the hematopoiesis by recruiting the progenitors from outside of the endocardium. Studies are required to unequivocally prove that endocardial cells undergo hematopoietic transition.”

- 3. The experimental design and therefore the results of the entire figure 5 are difficult to interpret. The ex vivo explant cultures is performed with tissues harvested from E8.5 embryos. An 18-day-long culture of heart explants in presence of the chemical Notch inhibition result in a decreased macrophage output compared to a vehicle-treated condition. Given the identification of Notch signaling as a critical regulator of Nkx2-5-dependent hematopoiesis, the authors use these results obtained with the chemical inhibition of Notch signaling as a validation of the platform. However, the timing of the genetic experiments show a full rescue at 9.5, which is compatible with a much earlier effect of Notch signaling. Hence, the results obtained by this ex vivo cultures raise the obvious question of the timing of Notch-dependence. When is then Notch needed for the Nkx2-5-dependent hematopoiesis? Is there a difference in Notch-dependency timing for setting up a cushion endocardial vs an hematopoietic program?*

The RA part of the experiment is also very difficult to interpret. On one end for the timing, as again an 18-day-long culture in presence of RA agonist and antagonist is used while in vivo there are already Dhhrs3⁺ CD41⁺ positive cells. As per Notch, when RA is important? For the specification of the hematopoietic program? For the emergence of CD41⁺ progenitors? For their differentiation into macrophages? Both Notch and RA are highly dynamic signaling, a better characterization of the timing of the requirement of these signaling is needed to support the claims made by the authors.

➡ We appreciate the Reviewer #3 for this insightful suggestion. Indeed, the long culture of explant colony assay makes the interpretation of the chemical interventions difficult. In the revised

manuscript, the chemicals (DAPT, atRA, DEAB) were added in three different regimens: ① during the OP9 co-culture alone (specification phase), ② during the methylcellulose culture (differentiation phase), and ③ during both OP9 co-culture and methylcellulose culture (the original regimen) (Figure 5A). While regimen ② gave mixed results, regimen ① recapitulates the original regimen ③, suggesting that the Notch and RA both regulate the hematopoietic specification phase.

Figure 5A

Figure 5B

4. Regarding the RA ex vivo studies, these are performed in presence of serum, which usually already contain retinol. Since the amount of atRA dictates the functional output (a recent Nat Cell Biol paper from the Sturgeon group show this elegantly in the context of hematopoietic development, and should probably be referenced as well), it would be important to test different doses of both the agonist atRA and antagonist DEAB. In addition, a proof that these small molecules are acting as expected (such as target genes expression) is totally lacking and very much needed.

➡ As the hematopoietic activities of the Caudal half and Heart are much lower than that of Yolk sac, the dose-dependency and the optimal doses of atRA and DEAB were determined by Yolk sac, and the optimal condition was applied to the Caudal half and the Heart. Both atRA and DEAB showed dose-dependent increase and decrease of the mRNA expression of RAR β , the target of RA signaling as in Nat Cell Biol paper from the Sturgeon group (Figure S7A). Based on this optimization experiment, 0.1 μ M atRA and 100 μ M DEAB were used to determine the colony number shown in Figure 5B.

Figure S7A

Minor comments:

1. Line 97 change “alienated” with “separated” or other synonym.

➡ Corrected.

2. In figure 5, many statistical analysis are missing, it is not clear whether they were not performed or whether the result are not significant.

➡ We have clarified the statistical analysis results in the revised manuscript. Thank you for your suggestion.

REVIEWERS' COMMENTS

Reviewer #1 (Remarks to the Author):

Summary: These researchers identified a regulatory network involved in the transition from endocardial cells to hematopoietic cells in the context of cardiac valve and septa remodeling. They analyzed single-cell RNA sequencing datasets and found that genes related to Notch and retinoic acid (RA) signaling were significantly downregulated in endocardial cells lacking Nkx2-5. Further in vivo and ex vivo analyses confirmed that the Nkx2-5-Notch axis is crucial for generating both hemogenic and cushion endocardial cells. Additionally, these authors discovered that the suppression of RA signaling through the expression of Dhhrs3 is important for the subsequent differentiation of these cells into macrophages. The researchers also conducted genetic ablation studies, which revealed that these macrophages are essential for cardiac valve remodeling. In conclusion, this study demonstrated that the Nkx2-5/Notch/RA signaling pathway plays a crucial role in the differentiation of macrophages from hematopoietic progenitors during the remodeling of cardiac valves.

The revised MS has improved the first version, although some criticisms about the Figures showing N1ICD immunofluorescence data are worth (Figure 2C). It is also surprising that in the response letter, the authors claim that: "The higher magnifications now enable readers to distinguish the cytosolic expression of N1ICD". I am not aware of any report showing N1ICD cytosolic expression in the wild type endocardium, it is always nuclear; the DAPI counterstaining should help to appreciate this. The N1ICD IFs shown in Figure 2C, although improved with respect to the first version, are not very convincing. E9.5 Wild type endocardial N1ICD expression is widespread, but it can only be observed in a few dispersed endocardial nuclei in Fig. 2C. In Nkx2-5 mutants, the two nuclei showing N1ICD expression are large and appear to be overlying the endocardium, or correspond to hematopoietic cells, perhaps the authors could comment on this. In the Methods section, there is no reference to the N1ICD antibody used, perhaps using thymamides to develop the signal would help. Also, an explanation of why two different N1ICD overexpressing mouse lines are used in this study would be useful to the dedicated reader

Reviewer #2 (Remarks to the Author):

The authors have addressed most comments and further improved the manuscript. Regarding Csf1r-ablation in the Nkx2-5 lineage (figure 6 and supplement), please provide information and quantification on macrophage ablation.

Reviewer #3 (Remarks to the Author):

This is the revised version of the manuscript by Liu et al.

The authors have done a very complete and documented job.

They have answered point by point to my questions and complete the manuscript by either performing new experiments or changing the text and I thank them very much for that.

Based on the elements mentions above and the answers to my questions, I am happy with the new version of the manuscript.

Reviewer #1:

The revised MS has improved the first version, although some criticisms about the Figures showing N1ICD immunofluorescence data are worth (Figure 2C). It is also surprising that in the response letter, the authors claim that: "The higher magnifications now enable readers to distinguish the cytosolic expression of N1ICD". I am not aware of any report showing N1ICD cytosolic expression in the wild type endocardium, it is always nuclear; the DAPI counterstaining should help to appreciate this. The N1ICD IFs shown in Figure 2C, although improved with respect to the first version, are not very convincing. E9.5 Wild type endocardial N1ICD expression is widespread, but it can only be observed in a few dispersed endocardial nuclei in Fig. 2C.

We appreciate the reviewer for the thorough review of the revised manuscript. N1ICD IFs was mostly expressed in the nucleus, as shown in the enlarged image below. The enlarged images below were added as Supplemental Figure 3. As the reviewer #1 pointed out, N1ICD was also partially detected in the cytoplasm. As shown in previous publications by Cui et al., *J Biol Chem.* 2004 and Liu et al., *Stem Cell Int.* 2021 (see images below), N1ICD was detected in the cytoplasm in other cell types. Therefore, we think it can be detected in the cytoplasm of endocardial cells as well.

This image is newly added as Supplementary Figure 3.

Images from
Cui et al., *J Biol Chem.* 2004.
 (PMID: 15082708)

Neural progenitor cells treated with two Notch1 ligand (NB-3 and Jagged1) and BSA as control. Nuclear translocation can be observed clearly, but some remained in the cytoplasm.

Images from **Liu et al., *Stem Cell Int.* 2021 (PMID: 33897781)**

Epithelial progenitor cells treated with Jagged-1 or DAPT. Arrowheads indicate cytoplasmic staining of NIICD. They used the same antibody of activated notch-1 as ours (Abcam #ab52301)

In Nkx2-5 mutants, the two nuclei showing NIICD expression are large and appear to be overlying the endocardium, or correspond to hematopoietic cells, perhaps the authors could comment on this.

The two NIICD⁺ cells in Nkx2-5 mutants co-expressed CD31, but it was weaker than that seen in other endocardial cells, suggesting that they are possibly hematopoietic endothelial cells. Given that Nkx2-5 mutants rarely have hematopoietic cell generation from endocardial cells in situ (Nakano et al, Nat Comm, 2013), it is possible that those two cells were the contamination of hematopoietic endothelial cells from circulation.

In the Methods section, there is no reference to the NIICD antibody used, perhaps using thyrמידes to develop the signal would help.

We apologize for the oversight. In the revised manuscript, we have added information on

the NIICD antibody in the Methods section.

Also, an explanation of why two different NIICD overexpressing mouse lines are used in this study would be useful to the dedicated reader.

These two lines are slightly different in that one is R26-knockin and the other is transgenic random integration. For unknown reason, R26-knockin version dies at around E10-11 and the transgenic version survives until late gestational stages, and the latter allows us to analyze the impact of NICD overexpression at later stages. It is now explained in the manuscript line 185~189 as follows. “Of note, *Nkx2-5^{cre/+}; R26^{+NIICD-GFP}* mice were underrepresented at around E10, and only 1 out of 16 E13.5 embryos were obtained. Therefore, we reproduced the above results using *Nkx2-5^{cre/+}; CAG-CAT-NIICD^{tg}* mice, which are not lethal at E13.5. *CAG-CAT-NIICD^{tg}* mice have been reported to show appropriate activity in the heart ²⁹, but gene introduction can occur randomly compared to the R26 strain, which may be the reason for the higher survival rate.”

Reviewer #2:

*The authors have addressed most comments and further improved the manuscript. Regarding *Csf1r*-ablation in the *Nkx2-5* lineage (figure 6 and supplement), please provide information and quantification on macrophage ablation.*

We compared macrophage presence in control and *Nkx2-5^{cre/+}; Csf1r^{fllox/fllox}* mutants at E13.5 and P10 by immunofluorescence/histochemical staining. The data did not show a statistically significant difference between Ctrl and Mut (Supplementary Figure 9). As previously reported (Shigeta et al., Dev Cell. 2019 (ref.13), Bain et al., Nat Commun. 2016 (ref 38), Epelman et al., Immunity. 2014 (ref39)), this is possibly due to the compensation from tissue macrophages of other origins.

This image is newly added as Supplementary Figure 9.

Reviewer #3:

This is the revised version of the manuscript by Liu et al.

The authors have done a very complete and documented job.

They have answered point by point to my questions and complete the manuscript by either performing new experiments or changing the text and I thank them very much for that.

Based on the elements mentions above and the answers to my questions, I am happy with the new version of the manuscript.

We thank the reviewer for helping us improve the manuscript.